



# Physical pedotransfer functions to compute saturated hydraulic conductivity from bimodal characteristic curves for a range of New Zealand soils

Joseph Aleander Paul Pollacco[1], Trevor Webb[1], Stephen McNeill[1], Wei Hu[2], Sam Carrick[1], Allan Hewitt[1], Linda Lilburne[1]

[1] Landcare Research, PO Box 69040, Lincoln 7640, New Zealand
[2] New Zealand Institute for Plant & Food Research Limited, Private Bag 4704, Christchurch 8140, New Zealand

*Correspondence to*: Joseph A.P. Pollacco (Pollacco.water@gmail.com)

**Abstract.** Descriptions of soil hydraulic properties, such as *soil moisture release curve*, $\theta(h)$, and *saturated hydraulic conductivities*, $K_s$, are a prerequisite for hydrological models. Since the measurement of $K_s$ is expensive, it is frequently derived from pedotransfer functions. Because it is usually more difficult to describe $K_s$ than $\theta(h)$ from pedotransfer functions, Pollacco et al. (2013) developed a physical unimodal model to compute $K_s$ solely from hydraulic parameters derived from the Kosugi $\theta(h)$. This unimodal $K_s$ model, which is based on a unimodal Kosugi soil pore-size distribution, was developed by combining the approach of Hagen-Poiseuille with Darcy's law and by introducing three tortuosity parameters. We report here on (1) the suitability of the Pollacco unimodal $K_s$ model to predict $K_s$ for a range of New Zealand soils, and (2) further adaptations to this model to adapt it to dual-porosity structural soils by computing the soil water flux through a continuous function of an improved bimodal pore-size distribution. The improved bimodal $K_s$ model was tested with a New Zealand data set derived from historical measurements of $K_s$ and $\theta(h)$ for a range of soils derived from sandstone and siltstone. The $K_s$ data were collected using a small core size of 100 mm, causing large uncertainty in replicate measurements. Predictions of $K_s$ were further improved by distinguishing topsoils from subsoil. Nevertheless, as expected stratifying the data with soil texture only slightly improved the predictions of the physical $K_s$ models because the $K_s$ model is based on pore-size distribution and the calibrated parameters were obtained within the physically feasible range. The improvements made to the unimodal $K_s$ model by using the new bimodal $K_s$ model are modest when compared to the unimodal model, which is explained by the poor accuracy of measured total porosity. Nevertheless, the new bimodal model provides an acceptable fit to the observed data. The study highlights the importance of improving $K_s$ measurements with larger cores.



**Keywords.** saturated hydraulic conductivity; bimodal; pedotransfer functions; Kosugi model; soil moisture release curves; Hagen-Poiseuille; tortuosity; soils; New Zealand; S-map


**Abbreviations. PTFs:** statistical pedotransfer functions; **PPTFs:** physically based pedotransfer functions; **S-map:** New Zealand soil database; $\theta(h)$ soil moisture release curve; $K_s$ saturated hydraulic conductivity




## 1 Introduction

Modelling of the water budget, irrigation, and nutrient and contaminant transport through the unsaturated zone requires accurate soil moisture release, $\theta(h)$, and unsaturated hydraulic conductivity, $K(\theta)$, curves. The considerable time and cost involved in measuring $\theta(h)$ and $K(\theta)$ directly for a range of soils mean that the information for specific soils of interest is often not available (Webb, 2003). Therefore, these curves are generally retrieved from pedotransfer functions (PTFs), which are statistical relationships that generate lower-precision estimates of physical properties of interest based on many rapid and inexpensive measurements (e.g., Balland and Pollacco, 2008; Pollacco, 2008; Anderson and Bouma, 1973; Webb, 2003).

The S-map database (Lilburne et al., 2012; Landcare Research, 2015) provides soil maps for the most intensively used land in New Zealand and is being gradually extended to give national coverage. S-map provides data for extensively used soil models, such as the soil nutrient model OVERSEER and the daily simulation model APSIM used by agricultural scientists. McNeill et al. (2012) used the New Zealand National Soils Database to derive PTFs to estimate $\theta(h)$ at five tensions from morphological data of soils mapped in S-map. One of the current weaknesses of S-map is a lack of capacity to estimate $K(\theta)$. Building on the work of Griffiths et al. (1999), Webb (2003) showed that morphologic descriptors for New Zealand soils can be used to predict $K_s$. However, the predictions of $K_s$ were found to be too coarse for application to the wide range of soils within S-map. Therefore, Cichota et al. (2013) tested published statistical PTFs developed in Europe and the USA to predict $\theta(h)$ and $K(\theta)$ for a range of New Zealand soils. They combined the best two or three PTFs to construct ensemble PTFs. They considered the ensemble PTF for $\theta(h)$ to be a reasonable fit, but the ensemble PTF for estimating $K_s$ exhibited large scatter and was not as reliable. The poor performance when estimating $K_s$ was possibly due to the absence of any measurements of pore-size distribution in their physical predictors (Watt and Griffiths, 1988; McKenzie and Jacquier, 1997), and also to the large uncertainties in the measurements from small cores (McKenzie and Cresswell, 2002 ; Anderson and Bouma, 1973). Consequently, there is an urgent need in New Zealand to develop a physically based pedotransfer function (PPTF) model for $K_s$ that is based on pore-size distribution.

Since PTFs developed to characterize $\theta(h)$ are more reliable than PTFs to characterize $K(\theta)$ (e.g., Balland and Pollacco, 2008; Cichota et al., 2013), Pollacco et al. (2013) developed a new class of physical pedotransfer function, PPTF, that predicts unimodal $K_s$ solely from hydraulic parameters derived from the Kosugi (1996) $\theta(h)$. The PPTFs are derived by combining the Hagen-Poiseuille and Darcy law and by incorporating three semi-empirical tortuosity parameters. The model is based on the soil pore-size distribution and has been successfully validated using the European HYPRES (Wösten et al., 1998; Wösten et al., 1999; Lilly et al., 2008) and the UNSODA databases (Leij et al., 1999; Schaap and van Genuchten, 2006), but has not yet been applied to New Zealand soils. Most New Zealand soils are considered to be structural, with two-stage drainage (Carrick et al., 2010; McLeod et al., 2008) and bimodal pore-size distribution (e.g. Durner, 1994). Romano and Nasta (2016) showed by using the HYDRUS-1D package that large errors arise in the computation of the the water fluxes if unimodal $\theta(h)$ and $K(\theta)$ are used in structural soils. We therefore propose to improve the unimodal Pollacco et al. (2013) $K_s$ PPTF model so that it can predict $K_s$ for structural soils with bimodal porosity.

Measured $K_s$ values exhibit notoriously high variability (Carrick, 2009). The variability is expected to increase as the sampling diameter decreases because small cores provide an unrealistic representation of the abundance and connectivity of macropores (McKenzie and Cresswell, 2002; Anderson and Bouma, 1973). McKenzie and Cresswell (2002) suggest that laboratory measurements should use cores with minimum diameter and length of 10–30 cm, with 25 cm diameter and 20 cm length the standard dimensions for Australian research. In New Zealand, $K_s$ has been obtained by using small cores,





commonly with 10 cm diameter and 7.5 cm length. This has contributed to very high variability in measured $K_s$ (Webb et al.,

78    2000).

The objectives of this research were to:
• test the suitability of the unimodal Pollacco et al. (2013) $K_s$ model to predict $K_s$ from New Zealand soils
• develop a $K_s$ bimodal model that makes predictions in structural soils solely from hydraulic parameters derived
from the Kosugi $\theta(h)$
• derive the uncertainties of the predictions of the $K_s$ bimodal model
• provide recommendations on the critical data sets that are required to improve the S-map database in New Zealand.
**2 Background**
**2.1 Kosugi unimodal characteristic and unsaturated hydraulic conductivity curve**
There are a number of closed-form unimodal expressions in the literature that compute the soil moisture release curve $\theta(h)$
and the unsaturated hydraulic conductivity $K(\theta)$ curves, such as the commonly used van Genuchten (1980) and Brooks and
Corey (1964) curves. We selected the physically based Kosugi (1996) closed-form unimodal log-normal function expression
of $\theta(h)$ and $K(\theta)$ because its parameters are theoretically sound and relate to the soil pore-size distribution (Hayashi et al.,
2009). Soils have a large variation in pore radius, $r$, which follows a log-normal probability density function. The unimodal
Kosugi log-normal probability density function of pore radius ($r$) is often written in the following form:
$$\frac{d\theta}{dr} = \frac{\theta_s - \theta_r}{r\,\sigma\,\sqrt{2\,\pi}}\exp\left\{-\frac{[\ln(r/r_m)]^2}{2\sigma^2}\right\} \tag{1}$$

where $\theta_r$ and $\theta_s$ [cm$^3$ cm$^{-3}$] are the *residual* and *saturated water contents*, respectively; $\ln(r_m)$ [cm] and $\sigma$ [-] are the mean and
variance of the log-transformed soil-pore radius, $\ln(r)$, respectively.

Let $S_e$ denote the effective saturation, defining $S_e(r) = (\theta - \theta_r)/(\theta_r - \theta_s)$, such that $0 \le S_e \le 1$. Integrating Eq. (1)
from 0 to $r$ yields the unimodal *characteristic curve* as a function of $r$:
$$S_e(r) = \frac{1}{2}erfc\left[\frac{\ln r_m - \ln r}{\sigma\sqrt{2}}\right] \tag{2a}$$

with $\quad r = \dfrac{r_m}{\exp\left[erfc^{-1}[2S_e]\sigma\sqrt{2}\right]} \tag{2b}$
where $erfc$ is the complementary error function.

The Young–Laplace capillary equation relates the soil-pore radius, $r$, to the equivalent *matric suction head*, $h$ (cm), at
which the pore is filled or drained (i.e., $r = Y/h$, where $Y = 0.149$ cm$^2$). Kosugi's unimodal *moisture release curve* $\theta_{uni}(h)$ can
be written in terms of $S_e$:
$$S_e(h) = \frac{1}{2}erfc\left[\frac{\ln h - \ln h_m}{\sigma\sqrt{2}}\right] \tag{3}$$

where $\ln(h_m)$ and $\sigma$ represent the mean and standard deviation of $\ln(h)$, respectively.





The unimodal Kosugi unsaturated hydraulic conductivity function $K(\theta)$ is written as:

$$K(S_e) = K_s \sqrt{S_e} \left\{ \frac{1}{2} \, erfc \left[ erfc^{-1}(2S_e) + \frac{\sigma}{\sqrt{2}} \right] \right\}^2 \qquad (4)$$

where $K_s$ (cm day$^{-1}$) is the saturated hydraulic conductivity.

$\theta_s$ is computed from the total porosity, $\phi$, which is deduced from *bulk density* ($\rho_b$) and *soil particle density* ($\rho_p$) as follows:

$$\phi = \left[ 1 - \frac{\rho_b}{\rho_p} \right] \qquad (5)$$

Due to air entrapment, $\theta_s$ seldom reaches saturation of the total pore space $\phi$ (Carrick et al., 2011). Therefore, to take into
account the fact that not all pores are connected, we perform the following correction of $\phi$ with $\alpha$ in the range [0.9, 1]:

$$\theta_s = \alpha \, \phi \qquad (6)$$

It is accepted that $\alpha = 0.95$ (Rogowski, 1971; Pollacco et al., 2013; Haverkamp et al., 2005; Leij et al., 2005), but in this
study the optimal $\alpha$ was found to be 0.98, since using a value of 0.95 resulted in several soil samples with $\theta_5$ ($\theta$ measured
at 5 kPa) greater than $\theta_s$, which is not physically plausible. This was due to the inaccuracy of measuring $\phi$ (discussed in
Sect. 4.1.2).
The feasible range of the Kosugi hydraulic parameters is summarized in Table 1. The $h_m$ and $\sigma$ feasible range is taken
from Pollacco et al. (2013), who combined data from the HYPRES (Wösten et al., 1998; Wösten et al., 1999; Lilly et al.,
2008) and UNSODA (Leij et al., 1999; Schaap and van Genuchten, 2006) databases.

**Table 1. please insert here**
**2.2 Pollacco unimodal saturated hydraulic conductivity model**
The *saturated hydraulic conductivity* model, $K_{s\_uni}$ (Pollacco et al., 2013) computes $K_s$ from the Kosugi parameters $\theta_s$, $\theta_r$, $\sigma$
and $h_m$ (or $r_m$). $K_{s\_uni}$ is based on the pore-size distribution (Eq. (1)) and the tortuosity of the pores. $K_{s\_uni}$ was derived by
adopting the method of Childs and Collisgeorge (1950) and modelling the soil water flux through a continuous function of
Kosugi (1996) pore-size distribution. This was performed by combining the Hagen-Poiseuille equation with Darcy's law and
introducing the connectivity and tortuosity parameters $\tau_1$, $\tau_2$ of Fatt and Dykstra (1951) and $\tau_3$ of Vervoort and Cattle (2003).
$K_{s\_uni}$ is computed as:

$$K_{s\_uni}(S_e) = C \, (1 - \tau_1) \, (\theta_s - \theta_r)^{\frac{1}{1-\tau_3}} \int_0^{S_e} r^{2(1-\tau_2)} dS_e \qquad (7)$$

with $\quad C = \dfrac{1}{8} \dfrac{\rho_w \, g}{\eta}$
where for water at 20°C, density of water $\rho_w = 0.998$ g cm$^{-3}$, acceleration due to gravity $g = 980.66$ cm s$^{-2}$, dynamic viscosity
of water $\eta = 0.0102$ g cm$^{-1}$ s$^{-1}$ and $C$ is a constant equal to $1.03663 \times 10^9$ cm day$^{-1}$.

Integrating with $S_e$ instead of $r$ avoids the complication of finding the minimum and maximum value of $r$. Isolating $r$ of
Eq. (2b) and replacing it in Eq. (7) gives:



$$K_{s\_uni}(S_e) = C (1-\tau_1) (\theta_s - \theta_r)^{\frac{1}{1-\tau_3}} \int_0^{S_e} \left\{ \frac{Y/h_m}{\exp\left[erfc^{-1}(2\,S_e)\,\sigma\sqrt{2}\right]} \right\}^{2(1-\tau_2)} dS_e \tag{8a}$$

$$\text{or } K_{s\_uni}(S_e) = C (1-\tau_1) (\theta_s - \theta_r)^{\frac{1}{1-\tau_3}} \int_0^{S_e} \left\{ \frac{r_m}{\exp\left[erfc^{-1}(2\,S_e)\,\sigma\sqrt{2}\right]} \right\}^{2(1-\tau_2)} dS_e \tag{8b}$$

and $\quad r_m = Y/h_m$

where $\tau_1$, $\tau_2$, $\tau_3$ are tortuosity parameters [0–1].


Note that $K_s = K_{s\_uni}(S_e = 1)$ (Eq. (8)). If tortuosity were not included ($\tau_1$, $\tau_2$, $\tau_3 = 0$), the pore-size distribution model
would mimic the permeability of a bundle of straight capillary tubes. Vervoort and Cattle (2003) state: "In reality soils are
much more complex, with twisted and crooked pores, dead-ending or connecting to other pores. This means that there is a
need to scale the permeability from the capillary tube model to include increased path length due to crookedness of the path
(tortuosity) or lack of connection between points in the soil (connectivity)". Soils that are poorly connected and have highly
crooked pathways theoretically have $\tau_1$, $\tau_2$, $\tau_3 \approx 0.9$. Further explanation of tortuosity is provided in Table 2.

**Table 2. Please insert here**

**2.3. Romano bimodal characteristic curve**
New Zealand soils are predominantly well structured, with two-stage drainage (Carrick et al., 2010; McLeod et al., 2008),
and therefore have a bimodal pore-size distribution (e.g. Durner, 1994). As $K_{s\_uni}$ is based on a unimodal curve, $\theta_{uni}(h)$, the
proposed bimodal model, $K_{s\_bim}$, should be based on a bimodal $\theta_{bim}(h)$ curve.

Borgesen et al. (2006) showed that structured soils have both *matrix* (inter-aggregate) pore spaces and *macropore*
(intra-aggregate) pore spaces. Thus, when the pores are initially saturated ($r > R_{mac}$) or ($h < H_{mac}$), the flow is considered
*macropore* flow, and when the soil is desaturated ($r < R_{mac}$) or ($h > H_{mac}$), the flow is considered *matrix flow*, as shown in
Fig. 1. $R_{mac}$ is the theoretical pore size $r$ that delimits macropore and matrix flow. To model bimodal pore-size distribution
Durner (1994) superposes two unimodal pore-size distributions by using an empirical weighting factor, $W$, which partitions
the volumetric percentage of macropore and matrix pores. Recently Romano et al. (2011) proposed the following Kosugi
bimodal $\theta_{bim\_rom}(h)$ distribution:
$$\theta_{bim\_rom}(h) = (\theta_s - \theta_r)\left\{ W\,erfc\left[\frac{\ln h - \ln h_{m\_mac}}{\sigma_{\_mac}\,\sqrt{2}}\right] + (1-W)\,erfc\left[\frac{\ln h - \ln h_m}{\sigma\,\sqrt{2}}\right] \right\} + \theta_r \tag{9}$$

where $\theta_s$, $\ln(h_{m\_mac})$ and $\sigma_{\_mac}$ are, respectively, the *saturated water content*, the *mean* and the *standard deviation* of $\ln(h)$ of
the macropore domain, $\theta_r$, $h_m$ and $\sigma$ are parameters of the matrix domain, and $W$ is a constant in the range [0,1).





### 3 Theoretical development of novel bimodal saturated hydraulic conductivity

We report on further adaptations to the physical model of Pollacco et al. (2013) to suit it to dual-porosity structural soils, which are common in New Zealand, solely from Kosugi hydraulic parameters describing $\theta(h)$. This involves:

- rewriting the Romano bimodal $\theta(h)$ (Sec. 3.1),
- developing a novel bimodal $K_s$ model based on the modified bimodal $\theta(h)$ (Sec. 3.2).

### 3.1 Modified Romano bimodal characteristic curve

We propose a modified version of $\theta_{bim\_rom}(h)$ (Eq. (9)) that does not use the empirical parameter $W$. Our modified function, $\theta_{bim}(h)$, is plotted in Fig. 1 and is computed as:

$$\theta_{bim}(h) = \theta_{bim\_mac}(h) + \theta_{bim\_mat}(h) \tag{10a}$$

$$\theta_{bim\_mac}(h) = \left[\theta_s - \theta_{s\_mac}\right] erfc\left[\frac{\ln h - \ln h_{m\_mac}}{\sigma_{\_mac}\sqrt{2}}\right] \tag{10b}$$

$$\theta_{bim\_mat}(h) = \left[\theta_{s\_mac} - \theta_r\right] erfc\left[\frac{\ln h - \ln h_m}{\sigma\sqrt{2}}\right] + \theta_r \tag{10c}$$

where $\theta_{s\_mac}$ is the *saturated water content* that theoretically differentiates *macropore* and *matrix* domains.

The shape of $\theta_{bim}(h)$ is identical to that of $\theta_{bim\_rom}(h)$, but the advantage of $\theta_{bim}(h)$ is that it uses the physical parameter $\theta_{s\_mac}$ instead of the empirical parameter $W$, and $\theta_{s\_mac}$ is more easily parameterized than $W$. $\theta_{s\_mac}$ is determined by fitting the hydraulic parameters $\theta_{s\_mac}$, $\theta_r$, $h_m$, $\sigma$ of $\theta_{bim\_mat}(h)$ (Eq. (10c)) solely in the matrix range ($r < R_{mac}$ or $h > H_{mac}$) by ensuring that $\theta_{s\_mac} < \theta_s$. Fig. 1 shows that $R_{mac}$ and $\theta_{s\_mac}$ delimit the matrix and the macropore domains and that $r_m$ of the Kosugi model is the inflection point of $\theta_{bim\_mat}(h)$ and $r_{m\_mac}$ is the inflection point of $\theta_{bim\_mac}(h)$.

**Fig. 1. Please put it here**

### 3.2 Novel bimodal saturated hydraulic conductivity model

Using $\theta_{bim}(h)$, we propose a new bimodal $K_{s\_bim}(S_e)$ that is derived following $K_{s\_uni}(S_e)$ (Eq. (7)) but for which we add a macropore domain:

$$K_{s\_bim}(S_e) = K_{s\_bim\_mat}(S_e) + K_{s\_bim\_mac}(S_e) \tag{11a}$$

$$K_{s\_bim\_mat}(S_e) = C \int_0^{S_e} \left[(1-\tau_1)\left(\theta_{s\_mac} - \theta_r\right)^{\frac{1}{1-\tau_3}} r_{matrix}^{2(1-\tau_2)}\right] dS_e \tag{11b}$$

$$K_{s\_bim\_mac}(S_e) = C \int_0^{S_e} \left[\left(1-\tau_{1_{\_mac}}\right)\left(\theta_s - \theta_{s\_mac}\right)^{\frac{1}{1-\tau_{3\_mac}}} r_{macropore}^{2\left(1-\tau_{2\_mac}\right)}\right] dS_e \tag{11c}$$

where $r_{macropore}$ is $r \geq R_{mac}$ and $r_{matrix}$ is $r < R_{mac}$.

The $r_{matrix}$ of Eq. (11b) is derived from Eq. (2b):

$$r_{matrix} = \frac{r_m}{\exp\left[erfc^{-1}[2\,S_e]\,\sigma\sqrt{2}\right]} \tag{12}$$





and $r_{\text{macropore}}$ is computed similarly as:
$$r_{macropore} = \frac{r_{m\_mac}}{\exp\left[erfc^{-1}\left[2\,S_e\right]\sigma_{\_mac}\sqrt{2}\right]} \tag{13}$$


We introduced $r_{\text{matrix}}$ (Eq. (12)) and $r_{\text{macropore}}$ (Eq. (13)) into $K_{\text{s\_bim}}$ (Eq. (11a)), giving the equation for $K_{\text{s\_bim}}$:
$$K_{s\_bim}(S_e) = C \int_0^{S_e} \left[ \begin{array}{l} \left(1-\tau_{1_{\_mac}}\right)\left(\theta_s - \theta_{s\_mac}\right)^{\frac{1}{1-\tau_{3\_mac}}} \left\{ \dfrac{r_{m\_mac}}{\exp\left[erfc^{-1}\left[2\,S_e\right]\sigma_{\_mac}\sqrt{2}\right]} \right\}^{2\left(1-\tau_{2\_mac}\right)} + \\[2em] \left(1-\tau_1\right)\left(\theta_{s\_mac} - \theta_r\right)^{\frac{1}{1-\tau_3}} \left\{ \dfrac{r_m}{\exp\left[erfc^{-1}\left[2\,S_e\right]\sigma\sqrt{2}\right]} \right\}^{2\left(1-\tau_2\right)} \end{array} \right] dS_e \tag{14a}$$

or
$$K_{s\_bim}(S_e) = C \int_0^{S_e} \left[ \begin{array}{l} \left(1-\tau_{1_{\_mac}}\right)\left(\theta_s - \theta_{s\_mac}\right)^{\frac{1}{1-\tau_{3\_mac}}} \left\{ \dfrac{\dfrac{Y}{h_{m\_mac}}}{\exp\left[erfc^{-1}\left(2\,S_e\right)\sigma_{\_mac}\sqrt{2}\right]} \right\}^{2\left(1-\tau_{2\_mac}\right)} + \\[2.5em] \left(1-\tau_1\right)\left(\theta_{s\_mac} - \theta_r\right)^{\frac{1}{1-\tau_3}} \left\{ \dfrac{\dfrac{Y}{h_m}}{\exp\left[erfc^{-1}\left(2\,S_e\right)\sigma\sqrt{2}\right]} \right\}^{2\left(1-\tau_2\right)} \end{array} \right] dS_e \tag{14b}$$


In Eq. (14b), $r_{\text{m\_mac}}$ is replaced by $Y/h_{\text{m\_mac}}$ and $r_{\text{m}}$ is replaced by $Y/h_{\text{m}}$ and for the computation of $K_s$ than $K_{\text{s\_bim}}$ ($S_e = 1$). Note
that the bimodal $K_s$ model requires that the flow in the macropore domain obeys the Buckingham–Darcy law. Therefore, this
model's performance may be restricted in cases of non-Darcy flow, such as non-laminar and turbulent flow, which may
occur in large macropores.

In this study $\sigma_{\_mac}$ is not derived from measured $\theta(h)$ because measured data in the macropore domain are difficult to
find, and so it will be treated as a fitting parameter. As discussed above, $\theta_{s\_mac}$, $\theta_r$, $\sigma$ and $h_m$ are optimized with $\theta_{uni}(h)$
measurement points only in the matrix range ($r < R_{\text{mac}}$ or h $> H_{\text{mac}}$), which means that $\theta_s$ is not included in the observation
data. In summary, $K_{\text{s\_bim}}$ requires optimization of the parameters $\tau_1$, $\tau_2$, $\tau_3$, and $\tau_{1\_mac}$, $\tau_{2\_mac}$, $\tau_{3\_mac}$ and $h_{m\_mac}$, $\sigma_{mac}$ (if no
data are available in the macropore domain). The theoretically feasible range of the parameters of $K_{\text{s\_bim}}$ is shown in Table 3.
**Table 3. Please put table here.**

One of the limitations of the New Zealand data set is that it has no $\theta(h)$ data points in the macropore domain. The
closest data point near saturation is $\theta(h = 50$ cm), which is in the matrix pore space. Carrick et al. (2010) found that $H_{mac}$
ranges from 5 to 15 cm, with an average $H_{mac} = 10$ cm, which corresponds to a circular pore radius of $R_{mac} = 0.0149$ cm (e.g.
Jarvis, 2007; Jarvis and Messing, 1995; Messing and Jarvis, 1993). Therefore, to reduce the number of optimized parameters
we make the following assumption:





$$h_{m\_mac} = \exp\left[\frac{\ln(H_{mac})}{2}\right] \qquad (15)$$

To illustrate this, the equivalent $r_{m\_mac}$ ($h_{m\_mac}$) point is shown in Fig. 1, where $r_{m\_mac}$ is the inflection point of the macropore
domain. The value 2 was found to be optimal. Fig. 1 also shows that the matrix and the macropore domains meet at $R_{mac}$
($H_{mac}$).

## 4 Methods

### 4.1 Data

#### 4.1.1 Selecting soil samples from New Zealand Soils Database

The soils data used in this study were sourced from two data sets. The first data set (Canterbury Regional Study; Table
4) soils were derived from eight soils series on the post-glacial and glacial surfaces of the Canterbury Plains (Webb et al.,
2000). The soils varied from shallow, well-drained silt loam soils to deep, poorly drained clay loam soils. Each soil series
had nine profiles. Three horizons in each soil profile were sampled from deep soils (topsoil, horizon with slowest
permeability, and the main horizon between these) and two from shallow soils (topsoil and the main horizon above gravels).
Grab samples were taken for particle size analysis, a 5.5 cm diameter core was taken from the middle of each horizon for
moisture release analysis, and three 10 cm diameter cores were taken from the upper part of each horizon for hydraulic
conductivity analysis.
The second data set was derived from the Soil Water Assessment and Measurement Programme to physically
characterize key soils throughout New Zealand in the 1980s. Soils selected from this data set are listed by region in Table 4.
All soils selected were from soils formed from sediments derived from indurated sandstone rocks, because this is the most
common parent material for soils in New Zealand and has a reasonably representative number of soils analysed for physical
properties. Selection of horizons and core size was similar to the Canterbury regional study, except that more subsoil
horizons were sampled at some sites, cores for hydraulic conductivity were not sampled in the topsoil horizon, and four to
six cores for hydraulic conductivity were sampled in subsoils.

#### 4.1.2 Measuring characteristic curves and total porosity

Laboratory analysis for particle size followed Gradwell (1972). The soil moisture release curves were derived by using 55
mm diameter cores according to the methods of Gradwell (1972).
The total porosity, $\phi$, described in Eq. (5) contains uncertainties from the measurement methods, where $\phi$ is derived
from separate measurements of particle density and bulk density, rather than being directly measured. The uncertainty in $\phi$
measurements appeared to have reduced the demonstrated benefits of using $K_{s\_bim}$ instead of $K_{s\_uni}$, which strongly relies on
$\phi\,\alpha - \theta_{s\_mac}$ and may have caused the optimal $\alpha$ to be 0.98 and not the commonly accepted value of 0.95 (Rogowski, 1971;
Pollacco et al., 2013; Haverkamp et al., 2005; Leij et al., 2005).
**Table 4. Please put here**

#### 4.1.3 Measuring saturated hydraulic conductivity performed with problematic small cores

The $K_s$ data used were collected and processed at a time when the best field practices in New Zealand were still being
explored. $K_s$ was derived using constant-head Mariotte devices (10 mm head) from three to six cores (100 mm diameter and





7.5 cm thickness) for each horizon. The $\log_{10}$ scale value of the standard error of the replicates of the measurements is shown
in Fig. 2, which shows large uncertainty in the measurements (up to three orders of magnitude). This uncertainty is due to:
a) **measurements of $\theta(h)$ and $K_s$ being taken on different cores**, which caused some mismatch between $\theta(h)$ and $K_s$,

resulting in 16 outliers that negatively influenced the overall fit of the $K_s$ model having to be removed from the data set

b) **side leakage** of some cores, which led to $K_s$ values that were too high (Carrick, 2009), resulting in six samples with

unusually high $K_s$ having to be removed from the data set

c) **misreporting low $K_s$** since the measurements of $K_s$ were halted when conductivity was less than 0.1 cm day$^{-1}$, resulting

in four samples with low $K_s$ having to be removed from the data set

d) **small core samples,** which led to considerable variability in the absence/presence of structural cracks caused by roots

or worm burrows (McKenzie and Cresswell, 2002; Anderson and Bouma, 1973) that were evident in dyed samples; we

therefore removed measured $K_s$ replicates that were too high and showed evidence of macropore abundance by having

values of $\theta_s - \theta_{s\_mac} > 0.05$.

We therefore selected 235/262 samples (90%) and removed only 27 outliers, which is minimal compared, for instance, to the
UNSODA (Leij et al., 1999; Schaap and van Genuchten, 2006) and HYPRES databases (Wösten et al., 1998; Wösten et al.,
1999; Lilly et al., 2008), which are used for the development of PTFs such as the ROSETTA PTF (Patil and Rajput, 2009;
Rubio, 2008; Young, 2009), and which were found to contain a large number of outliers. Using these databases, Pollacco et
al. (2013) selected only 73/318 soils (23%), which complied with strict selection criteria prior to modelling.

Note that the $K_s$ observations in the topsoils have greater variability than in the subsoil layers (Fig. 2). This is because
topsoils are more disturbed by tillage, planar fissures formed by wetting/drying, compaction, growth of plant roots and
earthworm burrowing. Therefore, the topsoils also have a greater abundance of macropores, and therefore are more prone to
error when the sampling is performed with a small core size that does not contain a representative volume of the macropore
network.

**Fig. 2. Please insert figure here**

**4.2 Inverse modelling**
The parameterization of the model was performed in two consecutive steps:
1. Optimization of $\theta_{s\_mac}$, $\theta_r$, $h_m$ and $\sigma$ of the unimodal Kosugi $\theta_{bim\_mat}(h)$ (Eq. (10c)) was performed by matching

observed and simulated $\theta(h)$ in the range $h < H_{mac}$ (as discussed, $\theta_s$ is not included in the observation data). The

feasible ranges of the Kosugi parameters are described in Table 1.

2. Optimization of the $\tau_1$, $\tau_2$, $\tau_3$ of the $K_{s\_uni}$ model (Eq. (8)) and $\tau_{1\_mac}$, $\tau_{2\_mac}$, $\tau_{3\_mac}$, $\sigma_{\_mac}$ parameters of the $K_{s\_bim}$

models (Eq. (14)), where the physical feasible ranges of the tortuosity parameters are described in Table 3.

The inverse modelling was performed using AMALGAM in MATLAB, which is a robust global optimization algorithm
(http://faculty.sites.uci.edu/jasper/sample/) (e.g., ter Braak and Vrugt, 2008). For each step we minimized the objective
functions described below.
**4.2.1 Inverting the Kosugi hydraulic parameters**
The objective function, $OF_\theta$, used to parameterize Kosugi's $\theta(h)$ at the following pressure points [5, 10, 20, 40, 50, 100,
1500 kPa], is described by:





$$OF_\theta = \sum_{i=1}^{i=N_\theta} \left[ \theta_{sim}(h_i, \mathbf{p}_\theta) - \theta_{obs}(h_i) \right]^{P_{ower}} \qquad (16)$$

where the subscripts *sim* and *obs* are simulated and observed, respectively. $P_\theta$ is the set of predicted parameters ($\theta_{s\_mac}$, $h_m$, $\sigma$)
and $P_{ower}$ is the power of the objective function.
The computation of $K_{s\_bim}$ requires $\theta(h)$ to be accurate near saturation, when the drainage is mostly from large pores,
and to achieve this we make $P_{ower}$ large (equal to 6).
**4.2.2 Calibrating the tortuosity parameters of the saturated hydraulic conductivity model**
The parameters of $K_{s\_uni}$ and $K_{s\_bim}$ models were optimized by minimizing the following objective function OF$_{ks}$:
$$OF_{ks} = \sum_{j=1}^{j=N_{ks}} \left[ \ln K_{s\_sim}(\mathbf{p_{ks}}) - \ln K_{s\_obs} \right]^2 \qquad (17)$$

where the subscripts *sim* and *obs* are simulated and observed, respectively. P$_{ks}$ is the vector of the unknown parameters. The
log transformation of OF$_{ks}$ puts more emphasis on the lower $K(\theta)$ and therefore reduces the bias towards larger conductivity
(e.g. van Genuchten et al., 1991; Pollacco et al., 2011). Also, the log transformation considers that the uncertainty in
measured unsaturated hydraulic conductivity increases as $K(\theta)$ increases.

The following transformation was necessary to scale the parameters to enable the global optimization to converge to a
solution:
$\tau_1 = 1 - 10^{-T1}$     (18)
where $T_1$ is a transformed tortuosity $\tau_1$.

Introducing Eq. (18) into $K_{s\_bim}$ Eq. (14) gives:
$$K_{s\_bim}(S_e) = C \int_0^{S_e} \left[ \begin{array}{c} 10^{-T_{1\_mac}} \left(\theta_s - \theta_{s\_mac}\right)^{\frac{1}{1-\tau_{3\_mac}}} \left\{ \dfrac{\dfrac{Y}{h_{m\_mac}}}{\exp\left[erfc^{-1}(2\,S_e)\,\sigma_{\_mac}\sqrt{2}\right]} \right\}^{2(1-\tau_{2\_mac})} + \\[3em] 10^{-T_1} \left(\theta_{s\_mac} - \theta_r\right)^{\frac{1}{1-\tau_3}} \left\{ \dfrac{\dfrac{Y}{h_m}}{\exp\left[erfc^{-1}(2\,S_e)\,\sigma\,\sqrt{2}\right]} \right\}^{2(1-\tau_2)} \end{array} \right] dS_e \qquad (19)$$

**5 Results and discussion**
We report on (1) the suitability of the $K_{s\_uni}$ model (European and American data sets, Pollacco et al., 2013) to predict $K_s$ for
New Zealand soils experiencing large uncertainties, as shown in Fig. 2; (2) improvements made by stratifying the data with
texture and topsoil/subsoil; and (3) improvements made by using the bimodal $K_{s\_bim}$ instead of the unimodal $K_{s\_uni}$. The
goodness of fit between simulated ($K_{s\_uni}$ or $K_{s\_bim}$) and observed $K_s$ was computed by the RMSElog$_{10}$:




$$RMSE_{\log10} = \sqrt{\frac{\sum_{j=1}^{j=N_{ks}} \left[\log_{10} K_{s\_sim} - \log_{10} K_{s\_obs}\right]^2}{N}}$$  (20)


where $N$ is the number of data points.

**5.1 Improvement made by stratifying with texture and topsoil/subsoil**

It was expected that stratifying with texture and topsoil/subsoil (layers) should improve the predictions of $K_s$ to only a
modest degree. This is because $K_{s\_bim}$ and $K_{s\_uni}$ are physically based models that are based on pore-size distribution, and
therefore stratifying with soil texture or topsoil/subsoil are not likely to provide extra information. For instance, Arya and
Paris (1981) showed that there is a strong relationship between pore-size distribution and the particle-size distribution and
therefore adding soil texture information should not improve the model.
**Table 5. please put table here**
As expected, no significant improvements were made by stratifying with soil texture compared with a model that
groups all texture classes (loam and clay) and layers (topsoil and subsoil) (overall improvement of 3%) (Table 5). However,
a significant improvement was made by stratifying by layer (topsoil and subsoil) (overall improvement of 23%), and
therefore the remaining results are presented by stratifying by layer. These results are obtained because topsoils have higher
macropores and a smaller tortuous path than that in subsoil, as demonstrated by $\tau_{1\_top} > \tau_{1\_sub}$ or $T_{1\_top} < T_{1\_sub}$, $\tau_{2\_top} > \tau_{2\_sub}$ ,
$\tau_{3\_top} > \tau_{3\_sub}$ (Table 6). It is important to note that tortuosity decreases as $\tau$ becomes closer to 1.
**Table 6. Please put table here**

**5.2 Improvement made by using $K_{s\_bim}$ instead of $K_{s\_uni}$**

Figure 3 shows an acceptable fit between $K_{s\_bim}$ and $K_{s\_obs}$ (RMSElog$_{10}$ = 0.450 cm day$^{-1}$), recognizing that the observations
contain large uncertainties since the measurements were taken by using small cores (Sect. 4.1.3). The overall improvement
made by using $K_{s\_bim}$ is somewhat modest (5% for all soils). As expected, the improvement is greater for topsoil containing
higher macroporosity (12% improvement) than for subsoil (4% improvement) (Table 6). This is because topsoil has higher
macropore $\theta_{mac}$ $(\theta_s - \theta_{s\_mac})$ (Table 7) caused by earthworm channels, fissures, roots and tillage than subsoil.
**Table 7. Please put table here**

The reason $K_{s\_bim}$ shows smaller-than-expected improvements compared to $K_{s\_uni}$ requires further investigation and
testing with a data set containing fewer uncertainties. One plausible explanation is that $K_{s\_bim}$ is highly sensitive to $\theta_s$,
computed from total porosity $\phi$ (Eq. (6)), which had inherent measurement uncertainties (Sect. 4.1.2). In addition, the
possible existence of non-Darcy flow in large biological pores may decrease the outperformance of the bimodal model over
the unimodal model.
**Fig. 3. Please insert Figure 3 here**

**5.2 Optimal tortuosity parameters**

The optimal tortuosity parameters of $K_{s\_bim}$ and $K_{s\_uni}$ (Table 6) show that the optimal parameters are within the physically
feasible limits, except for $\tau_{3\_mac}$ of the subsoil, which are greater than $\tau_3$. This is understandable because Pollacco et al.
(2013) found $\tau_3$ not to be a very sensitive parameter. As expected, $T_{1\_mac}$ is smaller than $T_1$ ($\tau_{1\_mac} > \tau_1$), which suggests that
the tortuosity parameters have a physical meaning.





The estimated value of the unimodal $T_1$ parameter $K_{s\_uni}$ derived from the UNSODA and HYPRES data sets ($T_1 = 0.1$)
(Pollacco et al., 2013) is very different from the value estimated in this present study ($T_1 = 6.5$). Cichota et al. (2013) also
reported that PTFs developed in Europe and the USA were not applicable to New Zealand. The reasons why these PTFs are
not directly applicable to New Zealand require further investigation.

**5.3 Uncertainty of the bimodal saturated hydraulic conductivity model predictions**

The practical application of the bimodal saturated hydraulic conductivity model, $K_{s\_bim}$, to New Zealand soils requires a
model for the uncertainty of the resultant predictions, since it is then possible to attach a value for the uncertainty of future
predictions of $K_s$. In a conventional parametric statistical model, the uncertainty model follows from the structure of the
fitting model itself. In the present work, $K_s$ is estimated using an inverse model and this has no associated functional
uncertainty model. For this reason, the uncertainty is derived empirically by fitting a relationship between the transformed
residuals of the model (the log-transformed measured $K_s$ minus the log-transformed estimated $K_s$) as a function of the
log-transformed estimated $K_s$. Although the uncertainty model could be derived from all the soils in the study, this results in
a pooled estimate for uncertainty (e.g., aggregated root mean square error). However, it has been observed that topsoils and
subsoils have different uncertainty behaviour for the estimated $K_s$, so it is desirable to include an indicator variable to
determine whether the soil is a topsoil or not. In explicit form,
$$\log_{10} K_{s\,obs} - \log_{10} K_{s\,sim} = a_1 L + a_0 + \epsilon \qquad (21)$$
where $a_0$ and $a_1$ are fitting constants, $L$ is an indicator variable specifying whether the soil is a topsoil (value 1), or a
subsoil (value 0), and $\epsilon$ is the uncertainty distribution. The distribution of the uncertainty $\epsilon$ could take a number of forms,
but there is no obvious choice, except that one might expect the distribution central measure to be unbiased. To avoid an
explicit distribution assumption, we fitted a conditional quantile model (Koenker, 2005) for the transformed residuals, based
on the $\tau$ quantile, where $\tau = 0.5$ corresponds to the conditional median, and $\tau = 0.025$ and $\tau = 0.975$ correspond
respectively to the 2.5% and 97.5% quantiles, and thus describe the 95% containment interval of the residuals.
The conditional quantile model Eq. (21) was fitted using $\tau = 0.5, 0.025$ and $0.975$ (Table 8). The results suggest a
strong dependence of the scale of the residuals on whether the soil is a topsoil or not, but the size of the 95% residual
containment interval is not dependent on the simulated $K_s$. Notably, the confidence interval for the fitted median ($\tau = 0.5$)
quantile model suggests that the uncertainty distribution median is unbiased; thus predictions from $K_{s\_bim}$ show no propensity
for bias, which is a desirable result.
**Table 8. Please put here**
Another way to illustrate the uncertainty model is to plot the observed $\log_{10} K_{s\,obs}$ against the estimated $\log K_{s\,bim}$,
with the fitted median, lower and upper 95% quantile lines, as shown in Fig. 4. The width of the 95% containment interval
for the residuals is narrower (i.e., the predictions appear to be more accurate) for topsoils. The quantile estimates for the
conditional median of both topsoil and subsoil are also shown in Fig. 4, with the shaded region showing the 95% confidence
interval of the median estimate. The shaded region covers the one-to-one line in Fig. 4, and thus there is no compelling
evidence that the median residual distribution is biased.
**Fig. 4. Please put here**





**5.4 Recommended future work to improve the New Zealand soil database**
A key outcome of this research will be to provide direction for future field studies to quantify soil water movement attributes
of New Zealand soils, and to prioritise which measurements will have the greatest value to reduce the uncertainty in
modelling of the soil moisture release and hydraulic conductivity relationships. Recommendations are:
•   Evaluate the spatial representativeness of the current soil physics data set and undertake more measurements of

hydraulic conductivity and soil water retention on key soils.

•   Use larger cores for measurements of hydraulic conductivity.
•   Take measurements of the moisture release curve and saturated hydraulic conductivity on the same sample.
•   Provide more accurate measurements of total porosity.
•   Conduct near saturation measurements of $\theta(h)$ and $K(\theta)$ to better characterize the macropore domain, which is

responsible for preferential flow behaviour.

•   Make more accurate measurements on slowly permeable soils ( $< 1$ cm day$^{-1}$), which are important for management

purposes but are not well represented in the current databases.

**7 Conclusions**
We report here on further adaptations to the saturated hydraulic conductivity model to suit it to dual-porosity structural soils
(Eq. 10) by computing the soil water flux through a continuous function of an improved Romano et al. (2011) $\theta(h)$ dual
pore-size distribution (Eq. 18). The shape of the improved Romano $\theta(h)$ distribution is identical to the improved $\theta(h)$, but the
advantage of the developed bimodal $\theta(h)$ is that it is more easily parameterized when no data are available in the macropore
domain.

The stratification of the data with texture only (loam or clay) slightly improved the predictions of the $K_s$ model, which

is based on pore-size distribution. This gives us confidence that the $K_s$ model is accounting for the effect of these physical
parameters on $K_s$. A significant improvement was made by separating topsoils from subsoils. The improvements are higher
for the topsoil, which has higher macroporosity caused by roots and tillage compared to subsoils. The reason why a model
with no stratification is not sufficient is unclear and requires further investigation.

The improvements made by using the developed bimodal $K_{s\_bim}$ (Eq. 18) compared to the unimodal $K_{s\_uni}$ (Eq. 8) is

modest overall, but, as expected, greater for topsoils having larger macroporosity. Nevertheless, an acceptable fit between
$K_{s\_bim}$ with $K_{s\_obs}$ was obtained when due recognition was given to the high variability in the measured data. We expect $K_{s\_bim}$
to provide greater improvement in $K_s$ predictions if more $\theta(h)$ measurements are made at tensions near saturation and if
measurements are made on larger cores and with more accurate measurements of porosity.
**Data availability**
The data are part of the New Zealand soil databases, available at http://smap.landcareresearch.co.nz/ and
https://soils.landcareresearch.co.nz/.





**Acknowledgements**
We are grateful to Leah Kearns and for Ray Prebble, who improved the readability of the manuscript. This project was
funded by Landcare Research core funding, through the New Zealand Ministry of Business, Innovation and Employment.

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

**Tables**
**Table 1. Feasible range of the Kosugi parameters and $\theta_5$ is $\theta$ measured at 5 kPa.**

|  | $\theta_s$ (cm$^3$ cm$^{-3}$) | $\theta_r$ (cm$^3$ cm$^{-3}$) | $\log_{10} h_m$ (cm) | $\sigma$ (-) |
|---|---|---|---|---|
| **Min** | $\theta_5$ | 0.0 | 1.23 | 0.8 |
| **Max** | 0.60 | 0.20 | 5.42 | 4.0 |







**Table 2. Description of the tortuosity parameters.**

| Tortuosity | Description |
| --- | --- |
| $\tau_1$ | Takes into account the increased path length due to crookedness of the path. When $\tau_1 = 0$ the flow path is perfectly straight down. When $\tau_1$ increases, the flow path is no longer straight but meanders. |
| $\tau_2$ | Theoretically represents the shape of a microscopic capillary tube. The $\tau_2$ parameter is used to estimate restrictions in flow rate due to variations in pore diameter and pore shape. When $\tau_2 = 0$ the shape of the capillary tube is perfectly cylindrical. When $\tau_2$ increases the tube becomes less perfectly cylindrical, which causes lower connectivity. |
| $\tau_3$ | High porosity soils tend to have large *effective pores*, $\theta_s - \theta_r$, which tend to be more connected than soils with smaller effective pores, which have more dead-ends. When $\tau_3 = 0$ the connectivity is the same between high and low porosity soils. When $\tau_3$ increases the connectivity of the soil increases (Vervoort and Cattle, 2003; Pollacco et al., 2013). Pollacco et al. (2013) found $\tau_3$ to be the least sensitive parameter. |



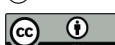



**Table 3. Theoretical constraints of the $K_{s\_bim}$ model.**

| Constraint | Explanation |
|---|---|
| $\theta_s > \theta_{s\_mac} \gg \theta_r$ | Self-explanatory. |
| $0 < \sigma_{mac} \leq 1.5$ | To avoid any unnecessary overlap of $\theta_{bim}$ with $\theta_{bim\_mat}$. |
| $1 > \tau_1 > \tau_{1\_mac} \geq 0$ | Flow in the macropore domain (larger pores) is expected to be straighter than in the matrix domain (smaller pores) due to reduced crookedness of the path. |
| $1 > \tau_2 > \tau_{2\_mac} \geq 0$ | It is expected that the shape of the 'microscopic capillary tube' of the macropore domain (larger pores) is more perfectly cylindrical than in the matrix domain (smaller pores). |
| $1 > \tau_3 > \tau_{3\_mac} \geq 0$ | The macropore domain has larger pores, and therefore it is assumed that the pores are better connected than the matrix pores. |





**Table 4. Soil series and classification.**

| Region | Soil series | No. of horizons | | New Zealand classification | Soil taxonomy |
|---|---|---|---|---|---|
| | | *Topsoils* | *Subsoils* | *Subgroup* | *Great group* |
| **Canterbury regional study** | Eyre | 6 | 8 | Weathered Orthic Recent | Haplustepts |
| | Templeton | 9 | 17 | Typic Immature Pallic | Haplustepts |
| | Wakanui | 9 | 17 | Mottled Immature Pallic | Humustepts |
| | Temuka | 9 | 16 | Typic Orthic Gley | Endoaquepts |
| | Lismore | 7 | 5 | Pallic Firm Brown | Dystrustepts |
| | Hatfield | 9 | 18 | Typic Immature Pallic | Humustepts |
| | Pahau | 9 | 18 | Mottled Argillic Pallic | Haplustalf |
| | Waterton | 9 | 15 | Argillic Orthic Gley | Endoaqualfs |
| **Canterbury** | Waimakariri | | 2 | Weathered Fluvial Recent | Haplustepts |
| | Lismore | | 1 | Pallic Orthic Brown | Dystrustepts |
| | Templeton | | 6 | Typic Immature Pallic | Haplustepts |
| | Wakanui | | 2 | Mottled Immature Pallic | Humustepts |
| | Temuka | | 2 | Typic Orthic Gley | Endoaquepts |
| **Manawatu** | Hautere | | 3 | Acidic Orthic Brown | Dystrudepts |
| | Levin | | 4 | Pedal Allophanic Brown | Humudepts |
| | Levin mottled | | 4 | Mottled Allophanic Brown | Humudepts |
| | Manawatu | | 1 | Weathered Orthic Recent | Haplustepts |
| | Paraha | | 3 | Mottled Immature Pallic | Haplustepts |
| | Westmere | | 2 | Typic Mafic Melanic | Humudepts |
| **Marlborough** | Brancott | | 3 | Mottled Fragic Pallic | Haplustepts |
| | Broadridge | | 3 | Mottled-argillic Fragic Pallic | Haplustalf |
| | Grovetown | | 3 | Typic Orthic Gley | Endoaquepts |
| | Raupara | | 1 | Typic Fluvial Recent | Ustifluvent |
| | Wairau | | 1 | Typic Fluvial Recent | Ustifluvent |
| | Woodburn | | 2 | Pedal Immature Pallic | Ustochrept |
| **Otago** | Dukes | | 1 | Typic Orthic Gley | Endoaquepts |
| | Linnburn | | 2 | Alkaline Immature Semiarid | Haplocambids |
| | Matau | | 4 | Typic Orthic Gley | Endoaquepts |
| | Otokia | | 1 | Mottled Fragic Pallic | Haplustepts |
| | Pinelheugh | | 2 | Pallic Firm Brown | Eutrudepts |
| | Ranfurly | | 2 | Mottled Argillic Semiarid | Haploargids |
| | Tawhiti | | 2 | Pallic Firm Brown | Eutrudepts |
| | Tima | | 2 | Typic Laminar Pallic | Haplustepts |
| | Waenga | | 2 | Typic Argillic Semiarid | Haploargids |
| | Wingatui | | 2 | Weathered Fluvial Recent | Haplustepts |
| **Southland** | Waikiwi | | 2 | Typic Firm Brown | Humudepts |
| | Waikoikoi | | 2 | Perch-gley Fragic Pallic | Fragiaqualfs |




**Table 5. Different combinations of texture, layer and RMSE$_{log10}$ reported by using $K_{s\_bim}$ and $K_{s\_uni}$ models.**

| Model form | RMSE$_{log10}$ | | |
|---|---|---|---|
| | $K_{s\_uni}$ | $K_{s\_bim}$ | $K_{s\_bim}$ - $K_{s\_uni}$ |
| **Model with combined texture and layer** | 0.583 | 0.560 | 0.023 |
| **Model with texture (loam and clay)** | 0.577 | 0.543 | 0.034 |
| **Model with topsoil and subsoil layers** | 0.450 | 0.430 | 0.020 |










**Table 6. Optimal tortuosity parameters of $K_{s\_uni}$ and $K_{s\_bim}$.**

|  |  | N | RMSE$_{log10}$ | $T_1$ | $\tau_2$ | $\tau_3$ | $T_{1\_mac}$ | $\tau_{2\_mac}$ | $\tau_{3\_mac}$ | $\sigma_{\_mac}$ |
|---|---|---|---|---|---|---|---|---|---|---|
| $K_{s\_bim}$ | **Topsoil** | 51 | 0.232 | 5.007 | 0.969 | 0.787 | 4.734 | 0.511 | 0.041 | 0.322 |
|  | **Subsoil** | 181 | 0.471 | 6.444 | 0.859 | 0.408 | 3.973 | 0.642 | 0.729 | 1.272 |
| $K_{s\_uni}$ | **Topsoil** | 51 | 0.259 | 5.859 | 0.967 | 0.530 | - | - | - | - |
|  | **Subsoil** | 181 | 0.491 | 6.484 | 0.854 | 0.316 | - | - | - | - |








**Table 7. Descriptive statistics of the optimized $\theta_{mac}$ ($\theta_s - \theta_{s\_mac}$), $\theta_s$, $h_m$ and $\sigma$ Kosugi hydraulic parameters. The bar represents the**
**average value, SD the standard deviation and $N$ the number of measurement points.**

| | N | $\overline{\theta_{mac}}$ | SD $\theta_{mac}$ | $\overline{\theta_s}$ | SD $\theta_s$ | $\overline{\theta_{s\_mac}}$ | SD $\theta_{s\_mac}$ | $\overline{lN\,h_m}$ | SD ln $h_m$ | $\overline{\sigma}$ | SD $\sigma$ | $\overline{K_s}$ | SD $K_s$ |
|---|---|---|---|---|---|---|---|---|---|---|---|---|---|
| | | (cm³ cm⁻³) | | (cm³ cm⁻³) | | (cm³ cm⁻³) | | (cm) | | (-) | | (cm h⁻¹) | |
| **Topsoil** | 51 | 0.038 | 0.035 | 0.48 | 0.04 | 0.45 | 0.04 | 6.43 | 1.02 | 3.00 | 0.61 | 167. | 101. |
| **Subsoil** | 181 | 0.030 | 0.030 | 0.42 | 0.05 | 0.39 | 0.06 | 5.39 | 1.66 | 2.64 | 0.86 | 19. | 42. |






**Table 8. Summary of the quantile regression fit of the log-transformed residuals.**

| Quantile | $a_0$ | | $a_1$ | |
|---|---|---|---|---|
| | **Estimate** | **95% CI** | **Estimate** | **95% CI** |
| $\tau = 0.025$ | −0.476 | $[-\infty, -0.44]$ | −0.574 | $[-0.62, \infty]$ |
| $\tau = 0.500$ | 0.041 | $[-0.036, 0.080]$ | 0.041 | $[-0.093, 0.053]$ |
| $\tau = 0.975$ | 0.357 | $[0.332, \infty]$ | 0.627 | $[-\infty, 0.711]$ |








**Figures**




**Figure 1. A typical Kosugi $\theta_{bim}$ (r) (Eq. (10b)) and $\theta_{bim\_mat}$ (r) (Eq. (10c)) with the matrix and macropore domains and the positions**
**of $\theta_s$ , $\theta_{s\_mac}$, $\theta_r$, $r_m$, $r_{m\_mac}$ , $R_{mac}$ shown.**






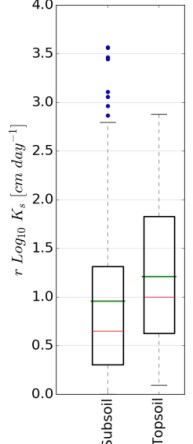

**Figure 2. Uncertainty of the standard error of the observed $K_s$ in topsoil and subsoil. The lines in the box show upper and lower quartiles, the median (red), and mean (green). Whiskers show values within 1.5 times the quartile spread; values outside this range are shown as plotted points.**






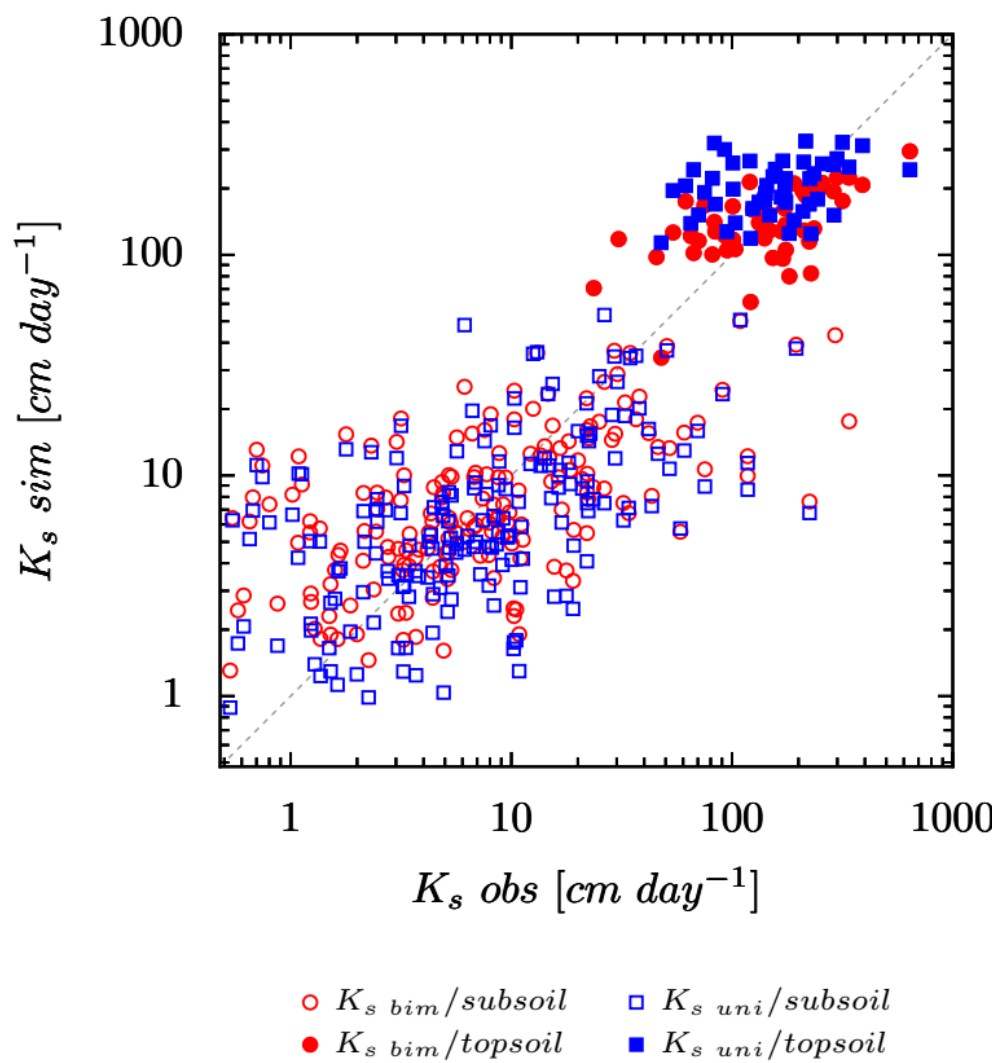


**Figure 3. Plot between $K_{s\_obs}$ against $K_{s\_bim}$ and $K_{s\_uni}$ for topsoil and subsoil.**



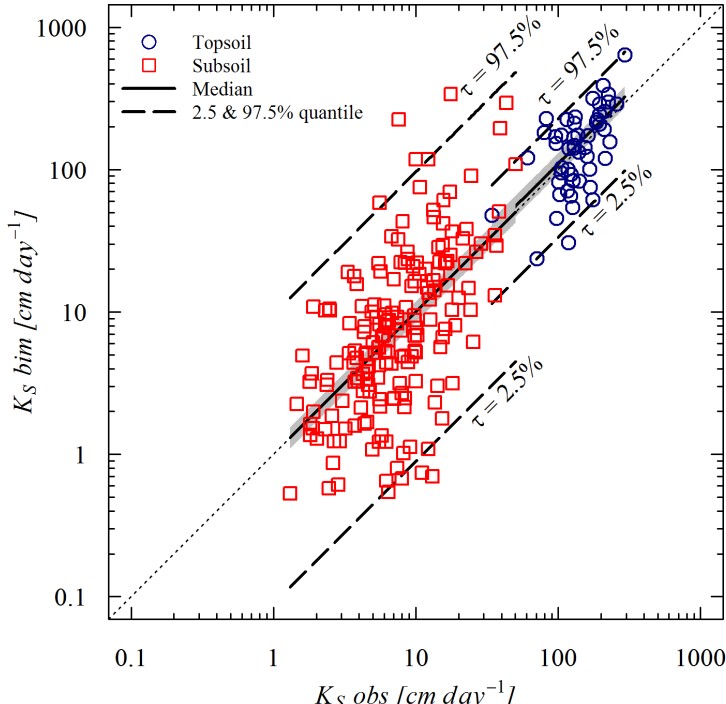



**Figure 4. Error of $K_{s\_bim}$ plotted against $K_{s\_obs}$ for topsoil and subsoil.**