# Peer review of "Physical pedotransfer functions to compute saturated hydraulic conductivity from bimodal characteristic curves for a range of New Zealand soils"

_Hydrology and Earth System Sciences, 2016_

## Referee Comment (RC1) · Anonymous Referee #1 · 13 Jan 2017

**Revision of Manuscript HESS-2016-636**

**Title: Physical pedotransfer functions to compute saturated hydraulic conductivity from bimodal characteristic curves for a range of New Zealand soils**

The authors propose a new mathematical formulation (inspired by the paper written by Pollacco et al., 2013) to estimate saturated hydraulic conductivity (Ks) from the parameters of the bimodal soil moisture release curve described by Romano et al. (2011). The authors use a data set of 235 soil samples collected in New Zealand and estimate Ks-values from unimodal (Pollacco et al., 2013) and bimodal (current manuscript, Pollacco et al., 2016) models. The authors observe improvements given by the bimodal model for topsoils that are affected by macroporosity. The evaluation of this manuscript is based on the following questions:

1) Is it a novel work based on a reliable scientific technique?
2) Is it clearly structured and well-written?
3) Are the experimental design and analysis of data adequate and appropriate to the investigation?

MAJOR COMMENTS

This scientific investigation can be considered novel since existing publications deal with Ks-values estimated from "unimodal" water retention curves. The data set is "robust" enough to satisfy the concluding remarks. However the manuscript is fragmented in too many small parts and requires some minor improvement in its structure.

This paper is potentially publishable since some of the material is of interest to the readership of HESS Journal. I have the following concerns on the current draft:

1) Subjective choice of $h_{m\_mac}$=3.16 cm (Eq. 15) in absence of measurements of data points near saturation. Maybe in this case, it would be recommended to optimize $h_{m\_mac}$ in order to increase objectivity and add flexibility.

2) The parameter W ("empirical" according to the authors) in the bimodal form of Romano et al. (2011) guarantees that the sum of the matrix and macropore domains gives $S_e$=1 (same role as in Durner, 1994). The authors replace it with a new parameter ($\theta_{s\_mac}$). Indeed they state that this new parameter is "physically sound" and can be easily optimized with the other soil moisture parameters in the matrix range delimited by $H_{mac}$, that is empirically fixed at 10 cm. Isn't it a contradiction? The authors should test this hypothesis on soil samples comprising measurements near saturation. This requires at least a few examples on soils taken from UNSODA or HYPRES for instance.

3) The RMSE-values obtained by this technique should be compared to the RMSE-values of existing methods (published in other articles) that estimate Ks from unimodal soil moisture parameters.

4) Experimental design needs to be clear: The authors mention that the water content values were measured at the following matric potential points: 5, 10, 20, 40, 50, 100, 1500 kPa (Lines 296-297)

please refer to the Book Methods of Soil Analysis, Part 4, Physical Methods" (J.H. Dane and G.C. Topp, eds.), pp. 692-698, SSSA Book Series N.5, Madison, WI, USA: which method was used to measure the moisture release curve? Hanging water column, suction tables, Pressure plate etc.

Overall I recommend minor revision of the manuscript with due attention to the above comments.

MINOR COMMENTS:

1) I doubt the term "pedotransfer function" is proper to identify the estimate of Ks from water retention parameters
2) Line 21 page 1: specify if you refer 100 mm to diameter or something else
3) Line 27 page 1: I agree that there are uncertainties related to the core sizes, but eventual improvements should be tested on larger cores.
4) Line 63 page 2: add references
5) Line 144 and 168, page 5: why is it [0,1]?
6) Lines 194-195, page 6: In Eqs 11b and 11c the two integral ranges are both Se=[0,1]. Shouldn't they be Se=[0 Se,mac] and [Se,mac,1]?
7) Lines 250-254, page 8: The determination of saturated water content (namely $\theta_s$) is rather easy, why do the authors use the artifact of Eq.6?
8) Fig. 2 page 25: improve overall quality, enlarge fonts
9) Fig. 3 page 26: please add the 1:1 line. Fig. 3 and 4 should be the same size
10) I encourage the authors to investigate on possible relationships between tortuosity parameters and soil physical parameters (texture, porosity etc)

---

## Referee Comment (RC2) · Anonymous Referee #2 · 27 Jan 2017

The intent of this paper is not very clear. On closer examination, even the title of the paper is problematic to me. My reasons are given below: 1. It is true that soil moisture release curve, theta(h), is still being measured in the laboratory despite being time-consuming. The hydraulic conductivity function K(h) is too expensive and time–consuming to measure and is typically reconstructed from the saturated hydraulic conductivity Ks and theta(h). Therefore what the authors seem to suggest in the paper is to use a bimodal theta(h) to compute Ks. The error involved will be too huge. In fact, it is common knowledge that an accurate K(h) can be obtained by measuring Ks and theta(h) rather than by estimating K(h) directly from theta(h). In fact, this is one

of the recommendations for future work in the paper. 2. Saturated Ks is not more time-consuming to measure compared to theta(h). 3. The approach chosen to determine Ks is strange as Ks depends on the voids in the soil. I can understand if one chooses the particle size distribution as providing the key parameters in a pedotransfer function to estimate Ks. Using theta(h) is an indirect process of getting the pore-size distribution but due to the time-consuming nature of the test, it is less suitable to be used as a proxy for pore-size distribution. 4. Even when using theta(h), it is expected that the matrix (micro) pores are the ones governing Ks but this is not evident from the paper. 5. The error for Ks shown in Figures 3 and 4, is about +/- one order. The errors in the measurement of Ks should be less despite the problems mentioned in Section 4.1.3. 6. Based on the above assessment, most of the equations presented in the paper have little value. In addition, none of the equations presented is a pedotransfer function in the traditional sense. 7. More relevant literature on estimating of saturated hydraulic conductivities should be cited e.g. Chapuis, R.P. (2004) Predicting the saturated hydraulic conductivity of sand and gravel using effective diameter and void ratio. Canadian Geotechnical Journal, 2004, 41:787-795, 10.1139/t04-022 Mbonimpa, M., Aubertin, M., Chapuis, R.P. (2002) Practical pedotransfer functions for estimating the saturated hydraulic conductivity. Geotechnical and Geological Engineering (2002) 20: 235. doi:10.1023/A:1016046214724

---

## Referee Comment (RC3) · Anonymous Referee #3 · 31 Jan 2017

Please find the following comments attached in pdf format to see special characters, subscripts, etc.

General comments The topic of the manuscript is very up to date, authors present a modified bimodal model to describe the water retention curve and couple it with a novel bimodal saturated hydraulic conductivity model. They consider models of Kosugi (1996), Romano et al. (2011) and Pollaco et al. (2013) in the model development. In most of the soil hydrological models bimodal characterization of soil hydraulic properties is not considered, although computation efficiency is enough advanced to allow to

improve that part of the models, therefore it is time to make a step forward in this field. It is very plausible that authors provide a soil physical explanations with the mathematical equations. I would recommend to consider modifying the title, because the presented method is closer to a model describing the Ks than to a PTF. For a PTF easily available soil properties are needed as input to calculate an unknown soil parameter. It is clear that at the end Ks is calculated from other parameters so it could be called as PTF, but in that case it would be very important to summarize what measured parameters are needed for the prediction and list the steps of calculations. If we call this procedure a PTF van Genuchten model could be called PTF as well because with that we can compute water content at any matric head (h), but for that we need data which is not easy/fast to measure. Therefore I would recommend to call these functions a type of "Ks fitting model", but even in that case it would be important to highlight which measured parameters are needed to calculate Ks. Further to this important issue the manuscript needs some minor revision prior to publication listed hereinafter.

Specific comments TEXT Line 11: here and in the entire text instead of "moisture release" "moisture retention" is more frequently used in the literature, therefore it might be more preferable to use. Line 18: here and in the entire text please use "structured soil" instead of "structural soil" if soil having aggregates is referred. Line 42: please refer to more recent PTFs. Line 94-95: it would be helpful for the reader to highlight what r and rm means. If rm refers to the mean of soil-pore radius I would suggest to write r with overbar. If $\sigma$ means the variance of the log-transformed soil-pore radius, please make it clear in the notation. Line 104: it might worth to give a number for the equation r=Y/h, than it is easier to refer it in 8b. Line 107: it might increase the readability/understanding if another notation would be used for the mean and standard deviation of ln(hm_mac). If first ln of hm_mac is calculated and then the mean and standard deviation of the transformed hm_mac, than the present notation does not tell it. Please check it. Lines 134-146: I hope that I didn't miss anything in Eq. 7-8b, if yes, sorry, just would like to clarify it. It seems that you have a small mistyping in the numbering of the equations, in line 146 you refer to Eq. 8 which is Eq. 7 in the text,

Eq. 8 is missing. Please correct it in the entire manuscript. If Se equals to 1 in Eq. 7 as mentioned in line 146, why is it included after Ks which in theory tells already that it is a saturated state because you use the notation "s"? If it is needed to follow the mathematical logic, a possible solution might be to add Se=1 under Eq. 7. If it is stated could Eq. 8a, 8b, 11a-14b, 19 be simplified? Lines 154 and 174: I would suggest to use "bimodal water retention curve" instead of "bimodal characteristic curve" to make it completely clear for the readers that you have to deal with both water retention curve ($\Theta(h)$) and hydraulic conductivity curve ($K(\Theta)$). Line 162: please give the terminology of Hmac too – as you did it for Rmac . Line 167: same as in line 107. Please check it. Line 226: maybe I miss something, for me it is not clear why 2 and why not 1.5. Can you please describe it? Line 235: Please describe shortly or rephrase what do you mean by main horizon? Lines 236, 237: in case of undisturbed samples please provide the volume of the core. Lines 247,248: please use cm also here. Line 251: please refer which method was used to measure particle density. Line 259: please use cm also here. Line 262: point a) does not fit into the uncertainty due to measurement error. It increases the error of the model, therefore better to mention it later when the performance of the bimodal model is analysed. Line 279-280: "anthropogenic disturbance and biological activity" might cover better the disturbances influencing soil porosity. Line 287: Eq. 10c is called "modified Romano bimodal" curve, why is it called unimodal Kosugi here? Line 290: please describe shortly how you optimized Ks_uni and Ks_bim models. Which measured parameters did you use? Line 302: could you provide reference or short explanation on why power was set to 6? Lines 307: instead of $K(\Theta)$ is not it more correct to write Ks? If yes, please rephrase sentence in lines 308-309. Line 321: please include if the difference is significant between unimodal and bimodal Ks models. Line 322-324: please include it in "materials and methods" section. Lines 319-322: it might worth to rephrase this section or include them separately under the subsections. Lines 326-330 and 332-335 are not totally in line, please harmonize them. Lines 341-344: is the improvement significant – overall or only in case of subsoils? Please include it in the text. Line 410: there is a mistyping, please

delete "improved" before "Romano ïĄś(h)". Please include the results of the modified bimodal model (10a) compared to Romano's model under "results" section too. Line 424: please include for what kind of soils you suggest to use the presented model and what are the limitations of its use.

TABLES Line 540: please rephrase, possible solution: "$\Theta_5$ which is". Why is $\Theta_5$ the minimum value of $\Theta_s$? Lines 545-546: "When ïĄť3 increases the connectivity of the soil increases", it seems to be in contradiction with lines 150-151 a 5th row of Table 3. Lines 555-558: please rephrase title of the table and its content because it is not clear in present from without reading the main text of the manuscript.

FIGURES Figure 3 and 4 has similar content, please consider to include them under 1 figure caption maybe including a) and b) figures.

Technical corrections Just a small suggestion, in Eq. 11a-11c and 12-13 maybe you can start with models regarding the macropore and then follow with the matrix similarly to Eq. 10a-10c, 14a-14b and 19, in this way you would have the same order in the equations in the entire manuscript. Please check Eq. 11a, 11b and 11c, because they have different size that other equations. Line 322: please put log10 in subscript.

Please also note the supplement to this comment:
http://www.hydrol-earth-syst-sci-discuss.net/hess-2016-636/hess-2016-636-RC3-supplement.pdf

---

## Editor Comment (EC1) · N. Romano (Editor) · 31 Jan 2017

Dear Authors, Your manuscript has received interesting comments from the discussants and now the deadline for discussion is approaching. To feed the important discussion step of this journal, I'd suggest you start providing preliminary responses to the comments received so far.

---

## Author Comment (AC1) · 16 Feb 2017

Dear Prof. Nunzio Romano,

We would like to express, our gratitude for your efforts for organizing the review of our article: Physical Pedotransfer Functions To Compute Saturated Hydraulic Conductivity From Bimodal Characteristic Curves For A Range Of New Zealand Soils. We really appreciate your positive evaluation, and the feedback that you find our research interesting and valuable. We also wish to acknowledge the time and quality of Reviewer 1 and 3. In the revised version of the paper we employed the following major modifications:

1) We changed the title of the manuscript from "Physical pedotransfer functions to compute saturated hydraulic conductivity from bimodal characteristic curves for a range of New Zealand soils" to "Saturated hydraulic conductivity model computed from bimodal water retention curves for a range of New Zealand soils" to reflect that the developed Ks model is not a pedotransfer function but a Ks model. We also made some few changes in the introduction to reflect the change of the title.

2) We rewrote section 4.1. Measurement of physical soil properties where we provided more emphasis on the measurement method and removed details of methods used to sample the data which did not add value to the paper.

3) Provided better explanation of the relationship between Hmac and $hm_{mac}(Eq.15)$.

4) Improved the quality of the equations.

Yours sincerely,

Joseph Alexander Paul Pollacco, Trevor Webb, Stephen McNeill, Wei Hu, Sam Carrick, Allan Hewitt, Linda Lilburne

Please also note the supplement to this comment:
http://www.hydrol-earth-syst-sci-discuss.net/hess-2016-636/hess-2016-636-AC1-supplement.pdf

---

## Author Comment (AC2) · 16 Feb 2017

**Physical Pedotransfer Functions To Compute Saturated Hydraulic Conductivity From Bimodal Characteristic Curves For A Range Of New Zealand Soils**

**Revision of Manuscript HESS-2016-636**

RESPONSE TO REFEREE 1

Dear Reviewer 1,

We would like to express, our gratitude for your efforts for your review of our article: *Saturated hydraulic conductivity model computed from bimodal water retention characteristic curves for a range of New Zealand soils*. We really appreciate your positive evaluation. We also wish to acknowledge for the time and the efforts of your comprehensive review that helped us to significantly improve the manuscript.

**MAJOR COMMENTS**

**However, the manuscript is fragmented in too many small parts and requires some minor improvement in its structure.**
  Thanks for bringing up this issue, to clarify the manuscript we simplified the subsections of the *Methods* section.

**1) Subjective choice of hm_mac=3.16 cm (Eq. 15) in absence of measurements of data points near saturation. Maybe in this case, it would be recommended to optimize hm_mac in order to increase objectivity and add flexibility.**
  We agree that Eq. 15 needs further explanation and therefore we rewrote the section as follow:

$$h_{m\_mac} \;=\; \exp\left[\frac{\ln\left(H_{mac}\right)}{P_{m\_mac}}\right] \qquad\qquad (15)$$

*where $P_{m\_mac}$ is a fitting parameter greater than 1. We found the fitted value of $P_{m\_mac}$ was 2.0, however this fitted parameter was very broadly determined. The cause might be that we are optimizing $\sigma_{\_mac}$ and therefore $h_{m\_mac}$ and $\sigma_{\_mac}$ might be linked. Linked parameters (Pollacco et al., 2008a, 2008b, 2009) means that there is an infinite combination of sets of linked parameters $h_{m\_mac}$ and $\sigma_{\_mac}$ which produces values of objective function close to that obtained with the optimal parameter set and for which there exists a continuous relationship between $h_{m\_mac}$ and $\sigma_{\_mac}$. Further research needs to determine if having more data in the macropore domain would reduce the cause of non-uniqueness. To illustrate $h_{m\_mac}$, the equivalent $r_{m\_mac}$ point is shown in Fig. 1, where $r_{m\_mac}$ is the inflection point of the macropore domain. Fig. 1 also shows that the matrix and the macropore domains meet at $R_{mac}$ ($H_{mac}$).*

**2) The parameter W ("empirical" according to the authors) in the bimodal form of Romano et al. (2011) guarantees that the sum of the matrix and macropore domains gives Se=1 (same role as in Durner, 1994). The authors replace it with a new parameter (θs_mac). Indeed they state that this new parameter is "physically sound" and can be easily optimized with the other soil moisture parameters in the matrix range delimited by Hmac, that is empirically fixed at 10 cm. Isn't it a contradiction? The authors should test this hypothesis on soil samples comprising measurements near saturation. This requires at least a few examples on soils taken from UNSODA or HYPRES for instance.**
  Thanks for your comments; nevertheless we do not fully understand why you believe there are contradictions. I improved the section and please inform us if it answers your concerns.

*The shape of $\theta_{bim}(h)$ is identical to that of $\theta_{bim\_rom}(h)$, but the advantage of $\theta_{bim}(h)$ is that it uses the physical parameter $\theta_{s\_mac}$ instead of the empirical parameter W, and $\theta_{s\_mac}$ ($\leq\theta_s$) is **more easily** parameterized than W* **particularly when there is no available data in the macropore domain**. **When we do not have data in the**

*macropore domain*, $\theta_{s\_mac}$ is determined by fitting the hydraulic parameters $\theta_{s\_mac}$, $\theta_r$, $h_m$, $\sigma$ of $\theta_{bim\_mat}(h)$ (Eq. (10b)) solely in the matrix range ($r < R_{mac}$ or $h > H_{mac}$) Fig. 1 shows that $R_{mac}$ and $\theta_{s\_mac}$ delimit the matrix and the macropore domains and that $r_m$ of the Kosugi model is the inflection point of $\theta_{bim\_mat}(h)$ and $r_{m\_mac}$ is the inflection point of $\theta_{bim\_mac}(h)$.

**3) The RMSE-values obtained by this technique should be compared to the RMSE-values of existing methods (published in other articles) that estimate Ks from unimodal soil moisture parameters.**

We agree that it will be best to compare our results with published data. We therefore compared our results to those of Pollacco et al., (2013):

*The RMSElog$_{10}$ of $K_{s\_uni}$ for subsoil is 0.47 cm day$^{-1}$ (Table 6) which is slightly worse compared to the RMSElog$_{10}$ of 0.420 cm day$^{-1}$ by using UNSODA and HYPRES data sets (Pollacco et al., 2013).*

**4) Experimental design needs to be clear: The authors mention that the water content values were measured at the following matric potential points: 5, 10, 20, 40, 50, 100, 1500 kPa (Lines 296-297) please refer to the Book Methods of Soil Analysis, Part 4, Physical Methods" (J.H. Dane and G.C. Topp, eds.), pp. 692-698, SSSA Book Series N.5, Madison, WI, USA: which method was used to measure the moisture release curve? Hanging water column, suction tables, Pressure plate etc.**

We agree that clarification of the experimental design is required and therefore we rewrote the section *4.1. Measurement of physical soil properties*

**MINOR COMMENTS:**

**1) I doubt the term "pedotransfer function" is proper to identify the estimate of Ks from water retention parameters**

We agree that the meaning of pedotransfer function is not well defined so therefore we are happy to change the title to:

*Saturated hydraulic conductivity model computed from bimodal water retention characteristic curves for a range of New Zealand soils*

We also made some minor corrections in the introduction to clarify that we are dealing with a model and not with a pedotransfer function.

**2) Line 21 page 1: specify if you refer 100 mm to diameter or something else**

Yes, we agree we need further specification.

*The $K_s$ data were collected using a small core size of 10 cm* **diameter.**

**3) Line 27 page 1: I agree that there are uncertainties related to the core sizes, but eventual improvements should be tested on larger cores.**

The manuscript purpose was to make the best out of the historical data stored in S-map https://smap.landcareresearch.co.nz/ which contains large uncertainties. Nevertheless, based on the recommendations made in section 6. *Recommended future work to improve the New Zealand soil database* we are now in the phase of collecting new data sets by using large core of size and by taking more measurement close to saturation and we plan to publish the results in due course.

**4) Line 63 page 2: add references**

We added the following historical reference since to our understanding there does not seem to be a specific paper written by Poiseuille which relates to the Hagen-Poiseuille and Darcy law.

Anon: The History of Poiseuille's Law, Annual Review of Fluid Mechanics, 25(1), 1–20, doi:10.1146/annurev.fl.25.010193.000245, 1993.

**5) Line 144 and 168, page 5: why is it [0,1)?**

This is a mathematical notation such that the feasible range is the internal interval including zero but not including one. This is because when $\tau_3 = 1$ then the $K_s$ model returns NaN.

**6) Lines 194-195, page 6: In Eqs 11b and 11c the two integral ranges are both Se=[0,1]. Shouldn't they be Se=[0 Se_mac] and [Se_mac,1]?**

The reviewer's question is to determine if in the matrix domain the integral should go from $[0, S_{e\_mac}]$ and in the macropore domain the integral should go to $[S_{e\_mac}, 1]$ compared to the two integrals evaluated over the interval $S_e=[0,1]$. This was questioned during the development of the model, and we decided to use the simplified notation for which the two integrals go to $S_e=[0,1]$ because numerically it makes little difference since the pore size distribution of the macropore and the matrix do not overlap since we constrained by $\sigma_{mac} < 1.5$.

**7) Lines 250-254, page 8: The determination of saturated water content (namely θs) is rather easy, why do the authors use the artefact of Eq.6?**

In the section 4.*1 Measurement of physical soil properties* we explained that the historical data used in this study had some issues in measuring $\theta$s, this is why we used Eq. 6:

*"The total porosity, $\phi$, described in Eq. (5) contains uncertainties from the measurement methods, where $\phi$ is derived from separate measurements of particle density and bulk density, rather than being directly measured".*

**8) Fig. 2 page 25: improve overall quality, enlarge fonts**

We improved the quality of Figure 2.

**9) Fig. 3 page 26: please add the 1:1 line. Fig. 3 and 4 should be the same size**

We added the 1:1 line in the caption. It is not possible that Fig.3 and Fig.4 are the same size since they were designed by using different software: Fig. 3 was designed by using PyX (Python) and Fig 4 by using *R*.

**10) I encourage the authors to investigate on possible relationships between tortuosity parameters and soil physical parameters (texture, porosity etc)**

We are currently collecting a new set of data where we are specifically measuring near saturation collected by using large cores, and we will investigate relationship between tortuosity parameters and other soil physical parameters.

---

## Author Comment (AC3) · 16 Feb 2017

**Physical Pedotransfer Functions To Compute Saturated Hydraulic Conductivity From Bimodal Characteristic Curves For A Range Of New Zealand Soils**

**Revision of Manuscript HESS-2016-636**

**RESPONSE TO REFEREE 2**

Dear Reviewer 2,

We would like to express, our gratitude for your efforts for your review of our article: *Saturated hydraulic conductivity model computed from bimodal water retention characteristic curves for a range of New Zealand soils*. We understand that you have concerns about the manuscript and we hope that we have addressed them.

**The intent of this paper is not very clear. On closer examination, even the title of the paper is problematic to me.**

We modified the title of the paper as suggested by reviewers 1 & 3 since they argue that the developed $K_s$ model is not a pedotransfer function but a functional model so therefore we changed the title to:

*Saturated hydraulic conductivity model computed from bimodal water retention characteristic curves for a range of New Zealand soils*

**1. It is true that soil moisture release curve, $\theta(h)$, is still being measured in the laboratory despite being time-consuming. The hydraulic conductivity function $K(h)$ is too expensive and time– consuming to measure and is typically reconstructed from the saturated hydraulic conductivity $K_s$ and $\theta(h)$. Therefore what the authors seem to suggest in the paper is to use a bimodal $\theta(h)$ to compute $K_s$. The error involved will be too huge. In fact, it is common knowledge that an accurate $K(h)$ can be obtained by measuring $K_s$ and $\theta(h)$ rather than by estimating $K(h)$ directly from $\theta(h)$. In fact, this is one C1 of the recommendations for future work in the paper. 2. Saturated $K_s$ is not more time-consuming to measure compared to $\theta(h)$.**

The reviewer raises an important issue. In some cases, $\theta(h)$ can be easier to measure, but our collective field and laboratory experience over many years is that the components of measurement required to estimate $K_s$ are more expensive to measure accurately, given the great variability we commonly expect for this property in New Zealand soils. We believe that this is due to the relatively young geomorphic development of the soils in this country. The purpose of this paper is to test one approach for modelling the $K(h)$ curve, which is an established valid approach in the scientific literature. You are correct that there are alternative approaches, which we will test in time as part of the S-map programme in NZ, but it is not the purpose of this paper to test these other options.

For clarification, among our *recommendations for future work* is a task to collect more accurate information on $\theta(h)$ and $K(\theta)$ to improve the development of $K_s$ models, which are required for the predictions of $K(\theta)$ to be fed into the S-map data base (https://smap.landcareresearch.co.nz/).

**3. The approach chosen to determine Ks is strange as Ks depends on the voids in the soil. I can understand if one chooses the particle size distribution as providing the key parameters in a pedotransfer function to estimate Ks. Using $\theta(h)$ is an indirect process of getting the pore-size distribution but due to the time-consuming nature of the test, it is less suitable to be used as a proxy for pore-size distribution.**

As mentioned above, the purpose of this paper is to test one approach for modelling the $K(h)$ curve, which is an established valid approach in the scientific literature. You are correct that there are alternative approaches, which we will test in time as part of the S-map programme in NZ, but it is not the purpose of this paper to test these other options.

Thanks for providing us with new insight in the development of $K_s$ model based on the particle size distribution. Nevertheless, the current S-map database does not have accurate particle size distribution, nevertheless we have good data on $\theta(h)$ and this is why we decided to use $\theta(h)$ to infer the pore size distribution based on Eq. 1. It will be interesting to compare $K_s$ models based on $\theta(h)$ and models based on particle size distribution for which, to my best of knowledge, a comparison has not yet been published.

*For instance, Arya and Paris (1981) showed that there is a strong relationship between pore-size distribution and the particle-size distribution and therefore adding soil texture information should not improve the model.*

**4. Even when using $\theta(h)$, it is expected that the matrix (micro) pores are the ones governing Ks but this is not evident from the paper.**

The percentage of pores contributing to macropores as discussed in the paper depends largely on $\theta_s$ - $\theta_{s\_mac}$.

**5. The error for Ks shown in Figures 3 and 4, is about +/- one order. The errors in the measurement of Ks should be less despite the problems mentioned in Section 4.1.3.**

This may suggest that the model is performing more accurately than the measured data! As mentioned in the paper high variability in Ks is widely recognised in the literature, which is why we are suggesting in the paper that future work use larger sample volumes (which is also shown in the literature to reduce measurement variability).

**6. Based on the above assessment, most of the equations presented in the paper have little value. In addition, none of the equations presented is a pedotransfer function in the traditional sense.**

As mentioned above, the purpose of this paper is to test one approach for modelling the $K(h)$ curve, which is an established valid approach in the scientific literature. As discussed, we changed the title of the paper to reflect our agreement that our $K_s$ model does not fit in the category of pedotransfer function but in the category of a model which is derived from principles of soil physics.

It is important to remember that the ultimate purpose of this paper is to derive $K_s$ from $\theta(h)$ data which is available in S-map, which is the national soil database of New Zealand. S-map addresses key issues that are important to New Zealand soils, although the methodology could in principle be applied elsewhere, and therefore the developed equations are useful to a greater or lesser extent outside New Zealand, depending on the soil information available other countries.

**7. More relevant literature on estimating of saturated hydraulic conductivities should be cited e.g.**

Many thanks for proposing literature to enrich our paper we included them in our paper. We found the following references to be highly relevant to this topic:

*Chapuis, R.P. (2004) Predicting the saturated hydraulic conductivity of sand and gravel using effective diameter and void ratio. Canadian Geotechnical Journal, 2004, 41:787-795, 10.1139/t04-022*

*Mbonimpa, M., Aubertin, M., Chapuis, R.P. (2002) Practical pedotransfer functions for estimating the saturated hydraulic conductivity. Geotechnical and Geological Engineering (2002) 20: 235. doi:10.1023/A:1016046214724*

---

## Author Comment (AC4) · 16 Feb 2017

**Physical Pedotransfer Functions To Compute Saturated Hydraulic Conductivity From Bimodal Characteristic Curves For A Range Of New Zealand Soils**

**Revision of Manuscript HESS-2016-636**

**RESPONSE TO REFEREE 3**

Dear Reviewer 3,

We would like to express, our gratitude for your efforts for your review of our article: *Saturated hydraulic conductivity model computed from bimodal water retention characteristic curves for a range of New Zealand soils*. We really appreciate your positive evaluation. We also wish to acknowledge for the time and the efforts of your comprehensive review that helped us to significantly improve the manuscript.

**I would recommend considering modifying the title, because the presented method is closer to a model describing the Ks than to a PTF.**
> We agree that the meaning of pedotransfer function is not well defined so therefore we are happy to change the title to:
>
> *Saturated hydraulic conductivity model computed from bimodal water retention characteristic curves for a range of New Zealand soils*
>
> We also made some minor corrections in the introduction to clarify that we are dealing with a functional model and not a pedotransfer function.

**TEXT**

**Line 11:** here and in the entire text instead of "moisture release" "moisture retention" is more frequently used in the literature, therefore it might be more preferable to use.
> We agree and are happy to systematically replace *moisture retention* in the manuscript with *moisture release*.

**Line 18:** here and in the entire text please use "structured soil" instead of "structural soil" if soil having aggregates is referred.
> We agree and are happy to systematically replace "*structured soil*" instead of "*structural soil*" in the manuscript where appropriate.

**Line 42:** please refer to more recent PTFs.
> We added a recent PTF reference developed in New Zealand:
> *Cichota, R., Vogeler, I., Snow, V. O., and Webb, T. H.: Ensemble pedotransfer functions to derive hydraulic properties for New Zealand soils, Soil Research, 51, 94–111, doi:10.1071/sr12338, 2013.*

**Line 94-95:** it would be helpful for the reader to highlight what r and $r_m$ means. If $r_m$ refers to the mean of soil-pore radius I would suggest writing r with overbar. If σ means the variance of the log transformed soil-pore radius, please make it clear in the notation.
> To clarify the meaning of the Kosugi parameters we rephrased Eq. 1 as follow:
>
> *where $\theta_r$ and $\theta_s$ [$cm^3 \ cm^{-3}$] are the residual and saturated water contents, $r_m$ [cm] is the median pore radius and σ [-] denotes the standard deviation of ln(r).*

**Line 104:** it might worth to give a number for the equation r=Y/h, than it is easier to refer it in 8b.

As in the recent paper *Using Bimodal Lognormal Functions to Describe Soil Hydraulic Properties* published by Romano et al., (2011) they did not include an equation number for the Young–Laplace capillary equation since it is understood that all soil scientist reading this specialized paper would be familiarised with the Young–Laplace capillary equation. Nevertheless, for clarity I added the following note in Eq. 8b:

I added

$r_m = Y/h_m$ (***Young–Laplace capillary equation***)

**Line 107:** it might increase the readability/understanding if another notation would be used for the mean and standard deviation of ln($h_{m\_mac}$). If first ln of $h_{m\_mac}$ is calculated and then the mean and standard deviation of the transformed $h_{m\_mac}$, than the present notation does not tell it. Please check it.

To clarify the meaning of the Kosugi parameters we rephrased Eq. 3 as follow:

*where $h_m$ [cm] is the median metric head*

**Lines 134-146:** I hope that I didn't miss anything in Eq. 7-8b, if yes, sorry, just would like to clarify it. It seems that you have a small mistyping in the numbering of the equations, in line 146 you refer to Eq. 8 which is Eq. 7 in the text, Eq. 8 is missing. Please correct it in the entire manuscript.

Thanks for noting this issue in the manuscript. We did not find any further issues of the numbering.

If $S_e$ equals to 1 in Eq. 7 as mentioned in line 146, why is it included after $K_s$ which in theory tells already that it is a saturated state because you use the notation "s"? If it is needed to follow the mathematical logic, a possible solution might be to add $S_e$=1 under Eq. 7. If it is stated could Eq. 8a, 8b, 11a-14b, 19 be simplified?

Just to clarify we defined in *2.1 Kosugi unimodal characteristic water retention and unsaturated hydraulic conductivity curve* $S_e(r) = (\theta - \theta_r)/(\theta_r - \theta_s)$.

We took on board your simplifications by rewriting Eq. 8, 11, 14 and 19 by integrating between 0 and 1 instead of 0 and $S_e$. I also simplified the notation for e.g. by replacing $K_{s\_bim}(S_e)$ to $K_{s\_bim}$. Nevertheless, we do not see how mathematically we can further simplify the equations.

**Lines 154 and 174:** I would suggest to use "bimodal water retention curve" instead of "bimodal characteristic curve" to make it completely clear for the readers that you have to deal with both water retention curve ($\Theta$(h)) and hydraulic conductivity curve (K($\Theta$)).

Thanks for improving the manuscript, we agree that replacing "*bimodal characteristic curve*" throughout the manuscript with "*bimodal water retention curve*" clarifies the meaning.

**Line 162:** please give the terminology of $H_{mac}$ too – as you did it for $R_{mac}$.

Thanks for helping us to clarify the manuscript, we made the modifications as suggested.

*$R_{mac}$ is the theoretical pore size r that delimits macropore and matrix flow **and $H_{mac}$ is the theoretical pressure that delimits macropore and matrix flow***

**Line 167:** same as in line 107. Please check it.

As suggested we made the following modifications.

*where $\theta_s$, $h_{m\_mac}$ and $\sigma_{\_mac}$ are, respectively, the saturated water content, **the median pore radius and the standard deviation of ln(h)** of the macropore domain, $\theta_r$, $h_m$ and $\sigma$ are parameters of the matrix domain, and W is a constant in the range [0,1).*

**Line 226:** maybe I miss something, for me it is not clear why 2 and why not 1.5. Can you please describe it?

We agree that Eq. 15 needs further explanation and therefore we rewrote the section as follow:

$$h_{m\_mac} = \exp\left[\frac{\ln\left(H_{mac}\right)}{P_{m\_mac}}\right] \qquad (15)$$

*where $P_{m\_mac}$ is a fitting parameter greater than 1. We found the fitted value of $P_{m\_mac}$ was 2.0, however this fitted parameter was very broadly determined. The cause might be that we are optimizing $\sigma_{\_mac}$ and therefore $h_{m\_mac}$ and $\sigma_{\_mac}$ might be linked. Linked parameters (Pollacco et al., 2008a, 2008b, 2009) means that there is an infinite combination of sets of linked parameters $h_{m\_mac}$ and $\sigma_{\_mac}$ which produces values of objective function close to that obtained with the optimal parameter set and for which there exists a continuous relationship between $h_{m\_mac}$ and $\sigma_{\_mac}$. Further research needs to determine if having more data in the macropore domain would reduce the cause of non-uniqueness. To illustrate $h_{m\_mac}$, the equivalent $r_{m\_mac}$point is shown in Fig. 1, where $r_{m\_mac}$ is the inflection point of the macropore domain. Fig. 1 also shows that the matrix and the macropore domains meet at $R_{mac}$ ($H_{mac}$).*

**Line 235:** Please describe shortly or rephrase what do you mean by main horizon?

We removed the following sampling description since it is confusing and it does not add extra clarifications to the results:

**

**Lines 236, 237:** in case of undisturbed samples please provide the volume of the core.

We rewrote the *4.1. Measurement of physical soil properties section* and we provided extra details of the field and laboratory methods

**Lines 247,248:** please use cm also here.

For consistency, we changed mm to cm in all the manuscript.

**Line 251:** please refer which method was used to measure particle density.

This was already mentioned in the manuscript:

*Laboratory analysis for particle size followed Gradwell (1972).*

**Line 259:** please use cm also here.

You are right for consistency with the rest of the text we changed mm to cm in all the manuscript.

**Line 262:** point a) does not fit into the uncertainty due to measurement error. It increases the error of the model, therefore better to mention it later when the performance of the bimodal model is analysed.

The variability in both $\theta$ and $Ks$ reflect variation within the stratum of a supposedly-uniform soil type. The effect is magnified by the small cores used, so in this sense it is an artefact of the measurement process and it is measurement error in the classical sense. We introduced this point in order to inform the reader concerning these historical datasets, which are considerably less accurate than modern datasets and that the reader should understand that if we had modern datasets the $K_s$ model should be much better.

**Line 279-280:** "anthropogenic disturbance and biological activity" might cover better the disturbances influencing soil porosity.

This is much better and concise; we implemented the corrections in to the manuscript.

**Line 287:** Eq. 10c is called "modified Romano bimodal" curve, why is it called unimodal Kosugi here?

We believe that we did not make any typos since Eq. 10 does not use the empirical weighting so it is no longer $\theta_{bim\_rom}(h)$.

**Line 290:** please describe shortly how you optimized $K_{s\_uni}$ and $K_{s\_bim}$ models. Which measured parameters did you use?

If I understand properly your comments, you wanted us to provided further explanation on the objective function which is described below.

*Optimization of the $\tau_1$, $\tau_2$, $\tau_3$ of the $K_{s\_uni}$ model (Eq. (8)) and $\tau_{1\_mac}$, $\tau_{2\_mac}$, $\tau_{3\_mac}$, $\sigma_{\_mac}$ parameters of the $K_{s\_bim}$ models (Eq. (14)), where the physical feasible ranges of the tortuosity parameters are described in Table 3.*

**Line 302:** could you provide reference or short explanation on why power was set to 6?
*The computation of $K_{s\_bim}$ requires $\theta(h)$ to be accurate near saturation, when the drainage is mostly from large pores, and to achieve **this balance we found by trial and error that best results are achieved** when $P_{ower} = 6$.*

**Lines 307:** instead of K(Θ) is not it more correct to write $K_s$? If yes, please rephrase sentence in lines 308-309.
*The log transformation of $OF_{ks}$ puts more emphasis on the lower $\boldsymbol{K_s}$ and therefore reduces the bias towards larger conductivity*

**Lines 319-322:** it might worth to rephrase this section or include them separately under the subsections.
For clarity we provide at the beginning of the *Result and discussion* section the plan of the layout of the results.

**Line 321:** please include if the difference is significant between unimodal and bimodal $K_s$ models.
We commented below in section 5.*2 Improvement made by using $K_{s\_bim}$ instead of $K_{s\_uni}$*

**Line 322-324:** please include it in "materials and methods" section
Thanks for your recommendation we moved the equation of goodness of fit into the *material and methods.*

**Lines 326-330 and 332-335** are not totally in line, please harmonize them.
If you are talking about the tabs than I lined them up. Thanks.

**Lines 341-344:** is the improvement significant – overall or only in case of subsoils? Please include it in the text.
The improvement is more significant in the topsoil than for the subsoil. We made a minor modification to the text to improve clarification:

*As expected, the **reasonable** improvement is greater for topsoil containing higher macroporosity (12% improvement) than for subsoil (4% improvement)*

**Line 410:** there is a mistyping, please delete "improved" before "Romano θ(h)". Please include the results of the modified bimodal model (10a) compared to Romano's model under "results" section too.
We agree that the wording was incorrect, we did not improve the model we just changed the form of parameterizing the model. Since the shape of the two models are identical we do not need to compare Romano θ(h) with θ(h) bimodal.

*We report here on further adaptations to the saturated hydraulic conductivity model to suit it to dual-porosity structured soils (Eq. 10) by computing the soil water flux through a continuous function of a **modified version** of Romano et al. (2011) θ(h) dual pore-size distribution (Eq. 18). The shape of the Romano θ(h) distribution is identical to the **modified** θ(h), but the advantage of the developed bimodal θ(h) is that it is more easily parameterized when no data are available in the macropore domain.*

**Line 424:** please include for what kind of soils you suggest to use the presented model and what are the limitations of its use.
This is indeed a valid question, but to answer this question correctly we would need to collect more soils samples in each subgroup (Table 4). Based on the section "*Recommended future work to improve New Zealand soil database*" we believe that the greatest challenge is to make predictions on slowly permeable soils as mentioned:

*Therefore, this model's performance may be restricted in cases of non-Darcy flow, such as non-laminar and turbulent flow, which may occur in large macropores.*

*Make more accurate measurements on slowly permeable soils ( < 1 cm day$^{-1}$), which are important for management purposes but are not well represented in the current databases.*

**TABLES**

**Line 540:** please rephrase, possible solution: "$\Theta_5$ which is". Why is $\Theta_5$ the minimum value of $\Theta_s$?

Due to uncertainties in measuring $\theta s$, we optimized $\theta_s$. The feasible range of $0.6 > \theta > \theta_5$. Since as mentioned *The closest data point near saturation is $\theta(h = 50\ cm)$, which is in the matrix pore space.*

**Lines 545-546:** "When τ3 increases the connectivity of the soil increases", it seems to be in contradiction with lines 150-151 a 5th row of Table 3.

We total agree with you this is why in section *Optimal tortuosity parameters* I commented on this contradiction:

*The optimal tortuosity parameters of $K_{s\_bim}$ and $K_{s\_uni}$ (Table 6) show that the optimal parameters are within the physically feasible limits, except for $\tau_{3\_mac}$ of the subsoil, which are greater than $\tau_3$. This is understandable because Pollacco et al. (2013) found $\tau_3$ not to be a very sensitive parameter.*

**Lines 555-558:** please rephrase title of the table and its content because it is not clear in present from without reading the main text of the manuscript.

We improved table 5 and the caption description.

**FIGURES**

Figure 3 and 4 has similar content, please consider including them under 1 figure caption maybe including a) and b) figures.

Thanks for suggesting merging figure 3 and figure 4. Since figure 3 relates to section *Improvement made by using Ks_bim instead of Ks_uni* and figure 4 relates to section *Uncertainty of the bimodal saturated hydraulic conductivity model predictions*, merging the 2 figures would give the wrong interpretation to the reader.

**Technical corrections**

Just a small suggestion, in Eq. 11a-11c and 12-13 maybe you can start with models regarding the macropore and then follow with the matrix similarly to Eq. 10a-10c, 14a-14b and 19, in this way you would have the same order in the equations in the entire manuscript.

Thanks for spotting this inconsistency, I will change the order of the equations mentioned starting from matrix than following for macropore. It is easier to start the explanations for matrix than for macropore.

Please check Eq. 11a, 11b and 11c, because they have different size that other equations.

Yes we corrected the quality of the equations and they now have the same size.

**Line 322:** please put log10 in subscript.

Thanks for spotting this typo:

---

## Author Comment (AC5) · 16 Feb 2017

Please find attached the revised manuscript

Please also note the supplement to this comment:
http://www.hydrol-earth-syst-sci-discuss.net/hess-2016-636/hess-2016-636-AC5-supplement.pdf

---

## Author Comment (AC6) · 16 Feb 2017

Please find attached the revised manuscript

Please also note the supplement to this comment:
http://www.hydrol-earth-syst-sci-discuss.net/hess-2016-636/hess-2016-636-AC6-supplement.pdf

---

## Author Comment (AC8) · 16 Feb 2017

Please find attached the revised manuscript.

Please also note the supplement to this comment:
http://www.hydrol-earth-syst-sci-discuss.net/hess-2016-636/hess-2016-636-AC8-supplement.pdf